# RNA-directed activation of cytoplasmic dynein-1 in reconstituted transport RNPs

Mark A McClintock[1], Carly I Dix[1†], Christopher M Johnson[2],
Stephen H McLaughlin[2], Rory J Maizels[1‡], Ha Thi Hoang[1§], Simon L Bullock[1*]

[1]Division of Cell Biology, MRC Laboratory of Molecular Biology, Cambridge, United Kingdom; [2]Division of Protein and Nucleic Acid Chemistry, MRC Laboratory of Molecular Biology, Cambridge, United Kingdom

**Abstract** Polarised mRNA transport is a prevalent mechanism for spatial control of protein synthesis. However, the composition of transported ribonucleoprotein particles (RNPs) and the regulation of their movement are poorly understood. We have reconstituted microtubule minus end-directed transport of mRNAs using purified components. A Bicaudal-D (BicD) adaptor protein and the RNA-binding protein Egalitarian (Egl) are sufficient for long-distance mRNA transport by the dynein motor and its accessory complex dynactin, thus defining a minimal transport-competent RNP. Unexpectedly, the RNA is required for robust activation of dynein motility. We show that a *cis*-acting RNA localisation signal promotes the interaction of Egl with BicD, which licenses the latter protein to recruit dynein and dynactin. Our data support a model for BicD activation based on RNA-induced occupancy of two Egl-binding sites on the BicD dimer. Scaffolding of adaptor protein assemblies by cargoes is an attractive mechanism for regulating intracellular transport.
DOI: https://doi.org/10.7554/eLife.36312.001

*For correspondence:
sbullock@mrc-lmb.cam.ac.uk

Present address: †AstraZeneca Discovery Sciences, Cambridge, United Kingdom; ‡MRC Laboratory of Molecular Cell Biology, University College London, London, United Kingdom; §MicroInventa Limited, London, United Kingdom

## Introduction

Targeting of mRNAs to specific locations within the cytoplasm can confer precise spatial control over protein synthesis and function (*Buxbaum et al., 2015*; *Holt and Schuman, 2013*; *Martin and Ephrussi, 2009*). By compartmentalising protein function, mRNA localisation contributes to diverse processes, including embryonic axis determination, epithelial polarity and neuronal plasticity. Trafficking of mRNAs frequently depends on the action of cytoskeletal motors, in particular those that move along the polarised microtubule network (*Mofatteh and Bullock, 2017*). However, the mechanisms by which specific mRNAs are recruited to, and transported by, microtubule motors remain unclear.

One of the most tractable systems for microtubule-based mRNA transport operates during early development of *Drosophila melanogaster* and is responsible for localising spatial determinants of embryonic patterning to microtubule minus ends. Transport of these mRNAs is dependent on the Egalitarian (Egl) and Bicaudal-D (BicD) proteins (*Bullock and Ish-Horowicz, 2001*), as well as the minus end-directed motor cytoplasmic dynein-1 (dynein) and its accessory complex dynactin (*Wilkie and Davis, 2001*). Egl is a 1004-amino-acid protein that directly associates with the specialised RNA stem-loops that mediate polarised transport (so-called RNA localisation signals) (*Dienstbier et al., 2009*). The basis of RNA recognition by Egl is not known, although an exonuclease-like domain between residues 557 and 726 is partly responsible (*Dienstbier et al., 2009*). Egl uses a short N-terminal region to bind BicD (*Dienstbier et al., 2009*), and C-terminal features to bind the LC8 dynein light chain (*Navarro et al., 2004*). Mammalian BicD orthologues – BICD1 and BICD2 – associate with dynein and dynactin (*Hoogenraad et al., 2001*). These observations have led to a model for linkage of localising mRNAs to the dynein transport machinery (*Figure 1A*). It is not

**eLife digest** In our cells, tiny molecular motors transport the components necessary for life's biological processes from one location to another. They do so by loading their cargo, and burning up chemical fuel to carry it along pathways made of filaments. For example, one such motor, called dynein, can move molecules of messenger RNA (mRNA) to specific locations within the cell. There, the mRNA will be used as a template to create proteins, which will operate at exactly the right place.

Transporting mRNA in this way is critical in processes such as embryonic development and the formation of memories; yet, this mechanism is still poorly understood. Previous work suggested that the mRNA is simply a passenger of the dynein motor, but McClintock et al. asked if this is really the case. Instead, could mRNA regulate its own sorting by controlling the activity of dynein?

Studying mRNA trafficking within the complex molecular environment of a cell is challenging, so mRNA transporting machinery was recreated in the laboratory. Only the proteins necessary to build a working system were included in the experiments. In addition to the filaments, the components included dynein and a complex of proteins known as dynactin, which allows the motor to move together with a protein called BICD2. A protein named Egalitarian was used to link the mRNA to BICD2.

By filming fluorescently labelled proteins and mRNAs, McClintock et al. discovered that mRNA strongly promotes the movement of the dynein motor. A structured section in the mRNA acts as a docking area for two copies of Egalitarian. This activates BICD2, which then binds to dynein and dynactin, thereby completing the transport machinery. According to these results, the mRNA directs the assembly of the system that will carry it within the cell.

Viruses such as HIV and herpesvirus hijack dynein motors to have their genetic information moved around a cell in order to propagate infection. Understanding precisely how mRNA is transported may help to develop new strategies to fight these viruses.

DOI: https://doi.org/10.7554/eLife.36312.002

known, however, if other factors co-operate with Egl and BicD to bridge mRNAs to the motor complex.

Another outstanding question is how the assembly of the transport complex, and the activity of the dynein motor within it, is controlled. Several lines of evidence indicate that BicD is a key player in these processes. By forming an extended coiled-coil homodimer, the isolated N-terminal region of mammalian BICD2 (BICD2N: containing coiled-coil domain 1 (CC1) and part of CC2) can bridge the interaction between dynein and dynactin, forming a mutually dependent triple complex (*Hoogenraad and Akhmanova, 2016*; *Splinter et al., 2012*; *Urnavicius et al., 2015*; *Zhang et al., 2017*). The binding of dynein to BICD2N and dynactin increases the incidence of processive movement dramatically (*McKenney et al., 2014*; *Schlager et al., 2014*), which is associated with repositioning of the dynein motor domains with respect to the microtubule (*Chowdhury et al., 2015*; *Zhang et al., 2017*). The motor also moves with higher velocity and has increased force output once bound to BICD2N and dynactin (*Belyy et al., 2016*; *McKenney et al., 2014*). The equivalent N-terminal region of *Drosophila* BicD stimulates dynein-based transport in vivo (*Dienstbier et al., 2009*), indicating that this mechanism is evolutionarily conserved.

Full-length BicD proteins interact poorly with dynein and dynactin and are therefore only weak activators of dynein motility (*Dienstbier et al., 2009*; *Hoogenraad et al., 2001*, *2003*; *Huynh and Vale, 2017*; *Liu et al., 2013*). While mechanistic details are still lacking, BicD appears to be autoinhibited by folding back of the third coiled-coil domain (CC3) onto the dynein-activating sequences in CC1/2 (*Dienstbier et al., 2009*; *Hoogenraad et al., 2001*; *Stuurman et al., 1999*). It has been proposed that the interaction of CC3 with cargo-binding proteins such as Egl or Rab6 (a G-protein that binds Golgi-derived vesicles) frees CC1/2 to interact with dynein and dynactin (*Dienstbier et al., 2009*; *Hoogenraad and Akhmanova, 2016*; *Hoogenraad et al., 2001*, *2003*; *Matanis et al., 2002*). Consistent with this model, mutating an essential residue in the shared Rab6- and Egl-binding site in CC3 prevents *Drosophila* BicD from associating with dynein in vivo (*Liu et al., 2013*).

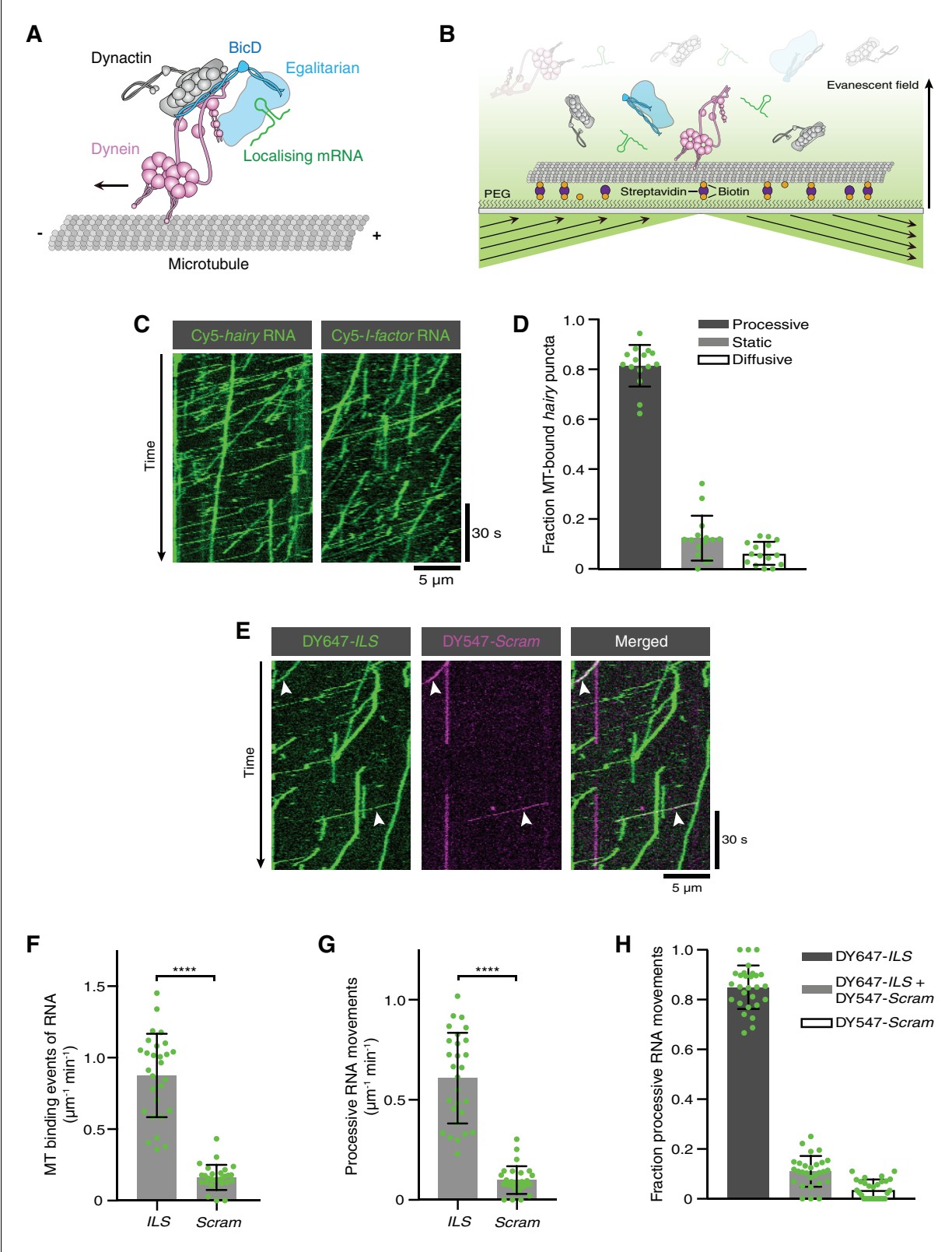

**Figure 1.** Reconstitution of dynein-based RNA transport with purified proteins. (**A**) Existing model for linkage of localising mRNAs to dynein. Note that there is no structural information available for Egl. (**B**) Diagram of TIRF-based in vitro motility assay. RNAs and proteins were incubated together for at least 1 hr on ice at the following molar concentrations: 100 nM dynein dimers, 200 nM dynactin, 100 nM Egl/BICD2 (with the operational assumption of two Egl molecules per BICD2 dimer) and 1 µM RNA. RNA-protein mixtures were typically diluted 40-fold and injected into imaging chambers

*Figure 1 continued on next page*

*Figure 1 continued*

containing microtubules that were pre-immobilised on passivated glass surfaces. (C) Examples of kymographs (time-distance plots) showing behaviour of Cyanine5 (Cy5)-labelled *hairy* or *I-factor* RNAs in the presence of Egl/BICD2, dynein and dynactin. Diagonal lines are processive movements. In these and other kymographs, the microtubule minus end is to the left. (D) Fraction of microtubule (MT)-associated *hairy* RNA complexes that are processive, static or diffusive. (E) Kymograph illustrating behaviour of DY647-labelled *ILS* and a scrambled (*Scram*) version of the sequence labelled with DY547 following co-incubation with Egl/BICD2, dynein and dynactin. Arrowheads: examples of co-transport of the two RNA species. (F and G) Numbers of RNA binding events on microtubules (F) and processive RNA movements (G) of *ILS* and *Scram* RNAs. (H) Fraction of processive RNA movements that contain signals from the *ILS* only, *Scram* only, or both RNAs. In (D) and (F-H), circles are values for individual microtubules. Error bars: SD. Statistical significance was evaluated with a Welch's *t*-test (F and G). ****p<0.0001.

DOI: https://doi.org/10.7554/eLife.36312.003

The following source data and figure supplements are available for figure 1:

**Source data 1.** Numerical values for plots presented in *Figure 1D,F–H*.
DOI: https://doi.org/10.7554/eLife.36312.009
**Figure supplement 1.** Alignment of the binding site for Rab6$^{GTP}$ and Egl in *Drosophila melanogaster* BicD with the corresponding region of mouse and human BICD2.
DOI: https://doi.org/10.7554/eLife.36312.004
**Figure supplement 2.** Supplemental data on the purification of Egl/BICD2.
DOI: https://doi.org/10.7554/eLife.36312.005
**Figure supplement 2—source data 1.** Numerical values for plots in *Figure 1—figure supplement 2C*.
DOI: https://doi.org/10.7554/eLife.36312.006
**Figure supplement 3.** Purity of protein preparations.
DOI: https://doi.org/10.7554/eLife.36312.007
**Figure supplement 4.** Kymographs illustrating additional behaviours of *hairy* RNA in the presence of Egl/BICD2, dynactin and dynein.
DOI: https://doi.org/10.7554/eLife.36312.008

It has recently been shown in vitro that the presence of Rab6 allows full-length BICD2 to associate with dynein and dynactin and thereby activate transport (*Huynh and Vale, 2017*). This observation provides direct evidence that association of a cargo-binding protein with CC3 stimulates the assembly of an active dynein-dynactin-BicD complex, although the stoichiometry of Rab6 and BICD2 in transport complexes was not investigated. Rab6 only associates with mammalian and *Drosophila* BicD proteins when it is GTP-bound (Rab6$^{GTP}$) (*Huynh and Vale, 2017*; *Liu et al., 2013*; *Matanis et al., 2002*; *Short et al., 2002*), a state induced by association with its target membranes (*Hutagalung and Novick, 2011*). These data suggest a mechanism for linking long-distance movement of dynein with the availability of a vesicular cargo. Egl, on the other hand, can associate with BicD CC3 in vitro in the absence of an RNA cargo (*Dienstbier et al., 2009*; *Liu et al., 2013*). This observation implies that the RNA is not involved in the relief of BicD autoinhibition by Egl, although this hypothesis has not been tested directly.

We set out to elucidate molecular mechanisms of dynein-based mRNA transport by Egl and BicD by reconstituting this process in vitro with purified components. Our results define a minimal set of proteins for RNA translocation on microtubules and show that the RNA strongly activates dynein motility. Stimulation of transport by RNA is not dependent on the Egl-LC8 interaction. Rather, our data support a model in which the RNA localisation signal overcomes BicD autoinhibition by augmenting the interaction of Egl with BicD CC3. Our study reveals a pivotal role of an RNA localisation signal in gating the activity of a microtubule motor, and give rise to a model in which cargoes stimulate dynein motility by scaffolding active adaptor protein assemblies.

## Results

### An in vitro assay for dynein-based mRNA transport

We set out to determine if purified dynein, dynactin, Egl and BicD are sufficient to induce mRNA transport in vitro. As no method is available for the purification of *Drosophila* dynein and dynactin, we established a system in which *Drosophila* Egl and an mRNA target are linked to mammalian dynein and dynactin complexes. We took advantage of the strong evolutionary conservation of the Egl/Rab6$^{GTP}$-binding site of BicD (*Figure 1—figure supplement 1*; [*Liu et al., 2013*]) to produce a complex of *Drosophila* Egl bound to mouse BICD2. This complex was purified from Sf9 insect cells

by co-expression of Egl with BICD2, as soluble Egl could not be recovered in the absence of its binding partner (*Figure 1—figure supplement 2A,B*). The Egl/BICD2 complex, which was captured using an affinity tag on Egl, was not associated with significant amounts of RNA (*Figure 1—figure supplement 2C*). This observation is consistent with previous evidence that RNA is not essential for the interaction of Egl with *Drosophila* BicD (*Dienstbier et al., 2009*; *Liu et al., 2013*). The 1.4 MDa human dynein complex and 1.1 MDa pig dynactin complex were purified from established recombinant and native sources, respectively (*Schlager et al., 2014*). The purity of these and other protein preparations used in the study is illustrated in *Figure 1—figure supplement 3*. RNAs were transcribed in vitro, and body-labelled by stochastic incorporation of fluorescent UTP.

Interactions of fluorescent RNA molecules with surface-immobilised microtubules were monitored by total internal reflection fluorescence (TIRF) microscopy in the presence of dynein, dynactin, and Egl/BICD2 (*Figure 1B*). RNAs and proteins were incubated together for at least 1 hr to promote complex assembly, followed by dilution to concentrations that allow discrimination of single molecules on microtubules. We first used the 3'UTR of the *hairy* mRNA, which mediates transport by a complex containing Egl, BicD, dynein and dynactin in the *Drosophila* embryo (*Bullock et al., 2003*; *Dix et al., 2013*). We observed frequent association of *hairy* RNA with microtubules in the imaging chamber (*Video 1*). Gratifyingly, 80% of microtubule-associated *hairy* RNA puncta underwent long-distance transport (*Figure 1C,D* and *Video 1*). As observed previously with a *Drosophila* extract-based system (*Soundararajan and Bullock, 2014*), *hairy* RNAs accumulated at microtubule minus ends following transport (*Figure 1—figure supplement 4A*) and were also capable of diffusive motion on the microtubule lattice (*Figure 1D* and *Figure 1—figure supplement 4B*). We also performed experiments with the *I-factor* retrotransposon RNA, which is transported in association with Egl, BicD, dynein and dynactin during *Drosophila* oogenesis (*Dienstbier et al., 2009*; *Dix et al., 2013*; *Van De Bor et al., 2005*). Like *hairy*, this RNA exhibited robust minus end-directed transport in our in vitro assay (*Figure 1C*). These experiments reveal that no additional proteins are required for microtubule-based mRNA transport in vitro.

To test if RNA localisation signals are selectively recognised in our assay conditions, we mixed the well-characterised 59-nucleotide (nt) Egl-binding element from the *I-factor* (*I-factor localisation signal* (*ILS*)) (*Dienstbier et al., 2009*; *Van De Bor et al., 2005*), which was labelled with DY647, with an equimolar amount of a scrambled version of the same sequence labelled with DY547. In the presence of Egl, BICD2, dynein, and dynactin, the *ILS* bound to microtubules ~five times more frequently than the mutant RNA and exhibited a similar relative increase in the number of processive movements (*Figure 1E–G*). These data reveal that the transport machinery retains selectivity for RNA localisation signals in our assay. Further analysis revealed that ~75% of the processive complexes that contained the scrambled RNA also had a signal from the *ILS* (*Figure 1E, H*), raising the possibility that much of the transport of the mutant RNA is an indirect consequence of association with active *ILS*-bound transport complexes.

## Egl/BICD2 and dynactin are required for mRNA transport by dynein

We next investigated the involvement of each of the protein complexes in the RNA transport process. We first used SNAP tags to fluorescently label dynein and either Egl or BICD2 in the Egl/BICD2 complex. Egl and BICD2 were co-transported with dynein and *hairy* RNA in the presence of dynactin (*Figure 2A* and *Figure 2—*

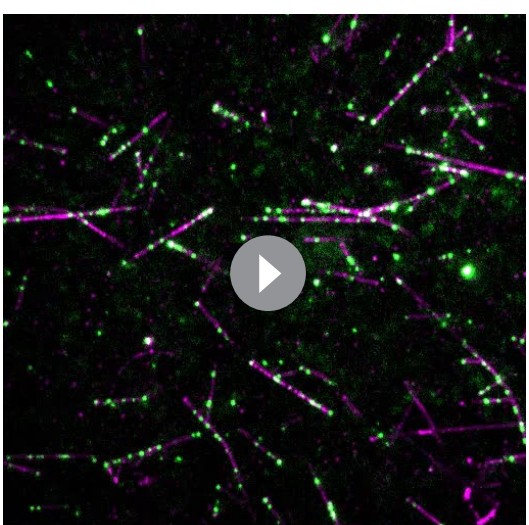

**Video 1.** Movements of Cy5-*hairy* RNAs on surface-immobilised microtubules in the presence of Egl/BICD2, dynactin and dynein. The RNA signal is shown in green. The position of the microtubules is indicated by a projection of the RNA signal over the course of the movie (magenta). Width of frame is 53.76 μm; movie corresponds to 252 s of real time.
DOI: https://doi.org/10.7554/eLife.36312.010

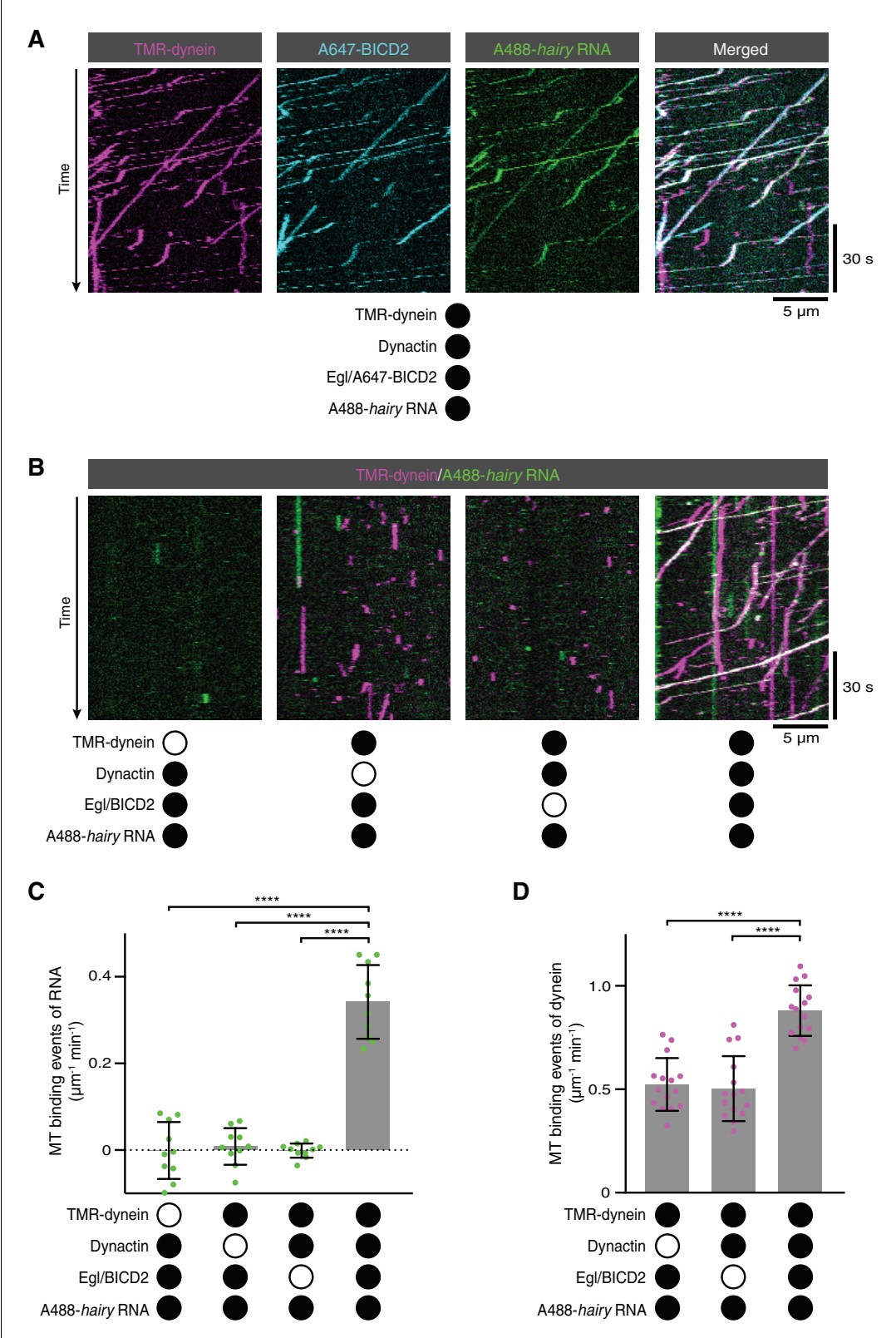

**Figure 2.** RNA transport by dynein requires the simultaneous presence of Egl/BICD2 and dynactin. (**A**) Kymographs showing co-transport of tetramethyrhodamine (TMR)-labelled dynein, Alexa647 (A647)-labelled BICD2 (included in the assembly as a complex with unlabelled Egl) and Alexa488 (A488)-labelled *hairy* mRNA in the presence of unlabelled dynactin. See *Figure 2—figure supplement 1* for equivalent experiment with Egl labelled in the Egl/BICD2 complex. (**B**) Kymographs illustrating the results of omitting dynein, dynactin or Egl/BICD2 from the assay. *Figure 2—figure*

*Figure 2 continued on next page*

*Figure 2 continued*

supplement 2 shows images of separate channels. (C) Binding of Cy5-*hairy* RNA to microtubules in the presence of the indicated proteins. Signals were corrected for background binding of RNA to the glass surface. (D) Binding of TMR-dynein to microtubules in the presence of the indicated proteins. Background correction was not necessary due to negligible association of dynein with the glass. In this and other figures, black or white circles indicate proteins that were present or absent from the experiment, respectively. In C and D, small circles are values for individual microtubules. Error bars: SD. Statistical significance was evaluated with an ANOVA test with Dunnett's multiple comparison correction. ****p<0.0001.
DOI: https://doi.org/10.7554/eLife.36312.011

The following source data and figure supplements are available for figure 2:

**Source data 1.** Numerical values for plots presented in *Figure 2C,D*.
DOI: https://doi.org/10.7554/eLife.36312.014
**Figure supplement 1.** Co-transport of Egl with dynein and RNA in the presence of BICD2 and dynactin.
DOI: https://doi.org/10.7554/eLife.36312.012
**Figure supplement 2.** Supplementary data on the requirement for Egl/BICD2 and dynactin for binding of RNA to microtubules and dynein motility.
DOI: https://doi.org/10.7554/eLife.36312.013

*figure supplement 1*; note that dynactin could not be labelled as it is from a native source). Next, we omitted individual protein complexes from the assembly mix. The association of *hairy* with microtubules was barely detected when Egl/BICD2, dynactin, or dynein was excluded (*Figure 2B,C* and *Figure 2—figure supplement 2*). Thus, the simultaneous presence of all three protein complexes is required to link RNA to microtubules. In the absence of Egl/BICD2 or dynactin, dynein rarely exhibited transport but could still associate with microtubules (*Figure 2B* and *Figure 2—figure supplement 2*). However, there was an ~two-fold increase in microtubule binding events when both Egl/BICD2 and dynactin were present (*Figure 2D*). Thus, the combination of Egl/BICD2 and dynactin stimulates dynein's ability to associate with microtubules and move processively in the presence of RNA.

## RNA-directed activation of dynein motility

As described in the Introduction, the prevailing model is that the association of Egl with BicD CC3 is sufficient to free the N-terminal region of BicD to interact with dynein and dynactin. Unlike Rab6, Egl can bind BicD in the absence of associated cargo, leading us to ask whether dynein and dynactin differentiate between RNA-bound and RNA-free Egl/BICD2. To address this question, we performed motility assays with Egl/BICD2, dynactin and fluorescent dynein in the presence and absence of RNA. Strikingly, the number of processive movements of dynein was ~six-fold higher when the RNA was present (*Figure 3A,B*). This reflected an increase in microtubule binding by dynein (*Figure 3C*), as well as the propensity for processive movement of those complexes associated with the microtubule (*Figure 3D*). The mean velocity and run length of dynein complexes bound to RNA were also significantly higher than those assayed in the absence of RNA (*Figure 3E,F*). We conclude that the RNA is required for robust stimulation of dynein motility and microtubule binding in the presence of Egl/BICD2 and dynactin.

We next asked if the RNA-directed activation of dynein was a consequence of the combination of Egl with a BicD protein from a different species by performing experiments with a preparation of *Drosophila* Egl and *Drosophila* BicD (DmBicD). The Egl/DmBicD complex was also produced by co-expression of both proteins in Sf9 insect cells and purification with an affinity tag on Egl. The *hairy* RNA significantly increased the number of processive movements of dynein in the presence of dynactin and Egl/DmBicD (*Figure 3G,H*). This effect was again associated with enhanced microtubule binding of the motor, as well as increased probability of processive movement after engaging with the microtubule (*Figure 3I,J*). As was observed in the experiments with Egl/BICD2, the RNA also enhanced the mean velocity and length of dynein movements (*Figure 3—figure supplement 1*). Thus, RNA also gates the activation of dynein motility by a co-evolved Egl/BicD complex.

## RNA promotes the assembly of the Egl/BicD/dynein/dynactin complex

We next considered two scenarios for how RNA stimulates dynein motility. First, the Egl/BicD/dynein/dynactin complex could be efficiently formed in the absence of RNA, with binding of RNA to Egl triggering a conformational change that activates processive dynein movement. Second, the ability of the Egl/BicD complex to interact with dynein and dynactin could be stimulated by the

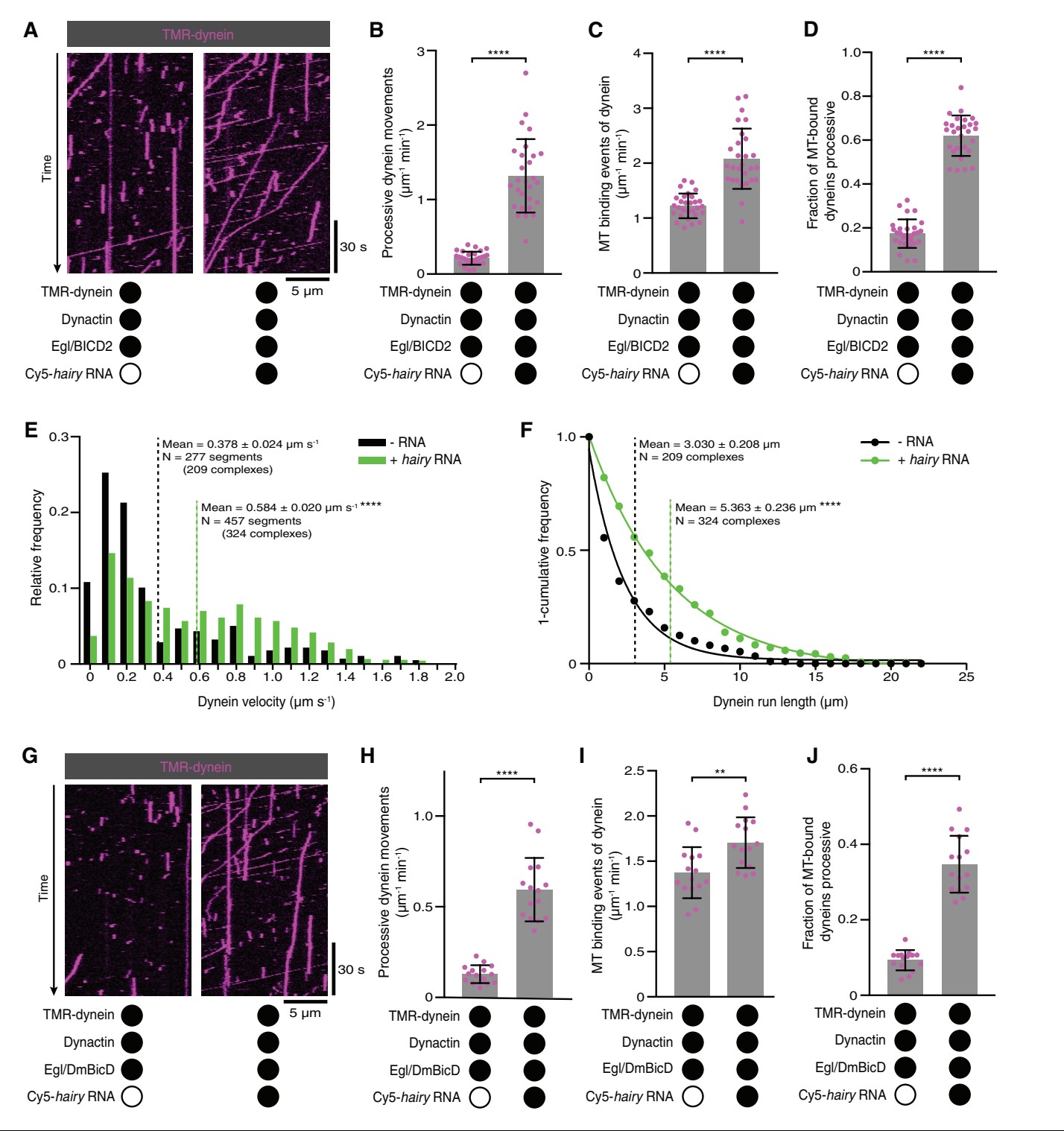

**Figure 3.** Activation of dynein motility by RNA. (**A**) Kymographs illustrating that *hairy* RNA increases the frequency of processive dynein movements in the presence of Egl/BICD2 and dynactin. (**B–D**) Numbers of processive dynein movements (**B**), microtubule-binding events of dynein (**C**) and fraction of microtubule-binding events that result in processive dynein movements (**D**) in the presence and absence of *hairy* RNA. (**E** and **F**) Distribution of segmental velocities (**E**) and run lengths (**F**) of dynein in the presence of Egl/BICD2 and dynactin ± *hairy* RNA (for experiments including *hairy* RNA, only those complexes associated with an RNA signal were analysed). (**G**) Kymographs illustrating that *hairy* RNA increases the frequency of processive dynein movements when dynactin and a complex of Egl bound to *Drosophila* BicD (DmBicD) is included in the assay. (**H–J**) Numbers of processive dynein movements (**H**), microtubule-binding events of dynein (**I**) and fraction of microtubule binding events that result in processive dynein movements

*Figure 3 continued on next page*

*Figure 3 continued*

(J) in the presence of dynactin and Egl/DmBicD ± *hairy* RNA. See **Figure 3—figure supplement 1** for velocity and run length distributions for these experiments. Errors: SD, except in E and F (SEM). In B-D, and H-J, circles are values for individual microtubules. In B, C, H, and J, statistical significance was evaluated with a Welch's *t*-test. In D and I, statistical significance was evaluated with a Student's *t*-test. In E and F, statistical significance (compared to the equivalent parameter in the absence of RNA) was evaluated with a Mann-Whitney test using raw, unfitted values. **p<0.01. ****p<0.0001.
DOI: https://doi.org/10.7554/eLife.36312.015

The following source data and figure supplements are available for figure 3:

**Source data 1.** Numerical values for plots in *Figure 3B–F,H–J*.
DOI: https://doi.org/10.7554/eLife.36312.018

**Figure supplement 1.** Supplementary data on dynein motility in the presence of dynactin and Egl/DmBicD ± *hairy* RNA.
DOI: https://doi.org/10.7554/eLife.36312.016

**Figure supplement 1—source data 1.** Numerical values for plots in *Figure 3—figure supplement 1A,B*.
DOI: https://doi.org/10.7554/eLife.36312.017

association of Egl with RNA, thus conferring different properties on the motor. To distinguish between these possibilities, we fluorescently labelled Egl in the purified Egl/BICD2 complex and monitored how *hairy* RNA affects its association with microtubule-bound dynein in the presence of dynactin. Although there was some association of dynein with Egl in the absence of RNA, the frequency of co-localisation increased by ~six-fold when the RNA was present (*Figure 4A–C*). We confirmed that the RNA also stimulates the association of BICD2 with microtubule-associated dynein by labelling BICD2 within the purified Egl/BICD2 complex (*Figure 4—figure supplement 1*). Many of the dynein complexes bound to Egl/BICD2 were motile (regardless of whether RNA was present of absent) (*Figure 4B*, *Figure 4—figure supplement 1* and *Figure 4—figure supplement 2*), indicating that they were also complexed with dynactin (*McKenney et al., 2014*; *Schlager et al., 2014*). Thus, the ability of RNA to activate processive dynein motion is associated with enhanced assembly of the Egl/BICD2/dynein/dynactin complex.

We next investigated if the assembly of the endogenous transport complex is stimulated by RNA. We immunoprecipitated a transgenically expressed GFP-tagged Egl protein from *Drosophila* embryo extracts in the presence and absence of exogenous *hairy* 3'UTR and assayed for co-precipitation of the p150 (DCTN1/Glued) subunit of dynactin and the heavy chain of dynein (Dhc) by western blotting (*Figure 4D,E*). p150 and Dhc were not detected in the Egl::GFP immunoprecipitate in the absence of exogenous RNA, indicating that the association of Egl with dynein and dynactin is of low affinity or low abundance. In contrast, the addition of the *hairy* RNA led to detectable co-precipitation of the dynein and dynactin components with Egl::GFP. Thus, assembly of the transport complex is promoted by the RNA in the context of both purified and endogenously-expressed proteins.

## The interaction of Egl with LC8 is not required for RNA-directed activation of dynein

The results described above raise the question of how RNA binding stimulates the association of Egl and BicD proteins with dynein and dynactin. We first asked if this involves the binding of Egl to the LC8 dynein light chain. Motility assays were performed with a purified Egl/BICD2 complex in which Egl has two mutations in a consensus LC8-binding site that abolish association with LC8 in vivo and in vitro (Egl[dlc2pt]; S965K + S969R) (*Navarro et al., 2004*). The Egl[dlc2pt]/BICD2 complex supported robust transport of *hairy* RNA in the presence of dynein and dynactin (*Figure 5A,B*). Moreover, the mutant Egl/BICD2 complex still supported the RNA-induced increase in processive movement and microtubule binding of dynein in the presence of dynactin (*Figure 5C–E*), as well as higher mean velocities and run lengths of the motor (*Figure 5—figure supplement 1*). Thus, the interaction of Egl with LC8 does not play a significant role in activation of dynein by RNA.

## The RNA localisation signal stabilises the Egl/BicD complex

These observations pointed to the other reported interaction of Egl/BICD2 with dynein and dynactin – that is the one mediated by BICD2N – as central to the activation of transport. As described in the Introduction, previous studies have indicated that occupancy of the Egl/Rab6[GTP]-binding site in BICD2 relieves autoinhibition, licensing BICD2N to interact with dynein and dynactin (*Huynh and Vale, 2017*; *Liu et al., 2013*). During handling of the purified Egl/BICD2 complex, we noticed that it

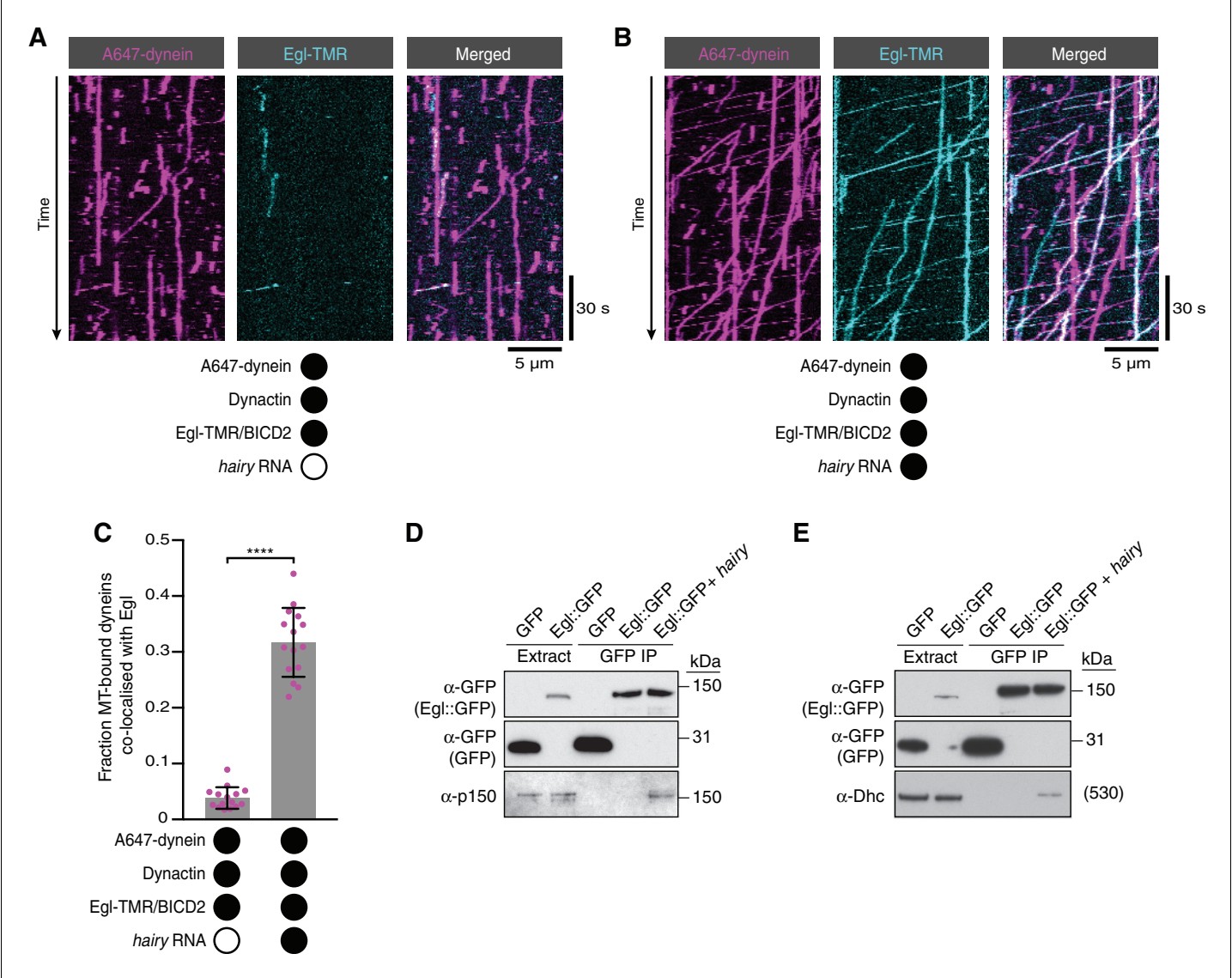

**Figure 4.** RNA stimulates the assembly of the transport complex. (**A** and **B**) Kymographs illustrating the behaviour of fluorescent dynein and Egl (included in the assembly in a complex with unlabelled BICD2) in the presence of dynactin ± *hairy* RNA. (**C**) Fraction of microtubule-bound dyneins that associate with Egl in the presence of dynactin ± *hairy* RNA. Circles are values for individual microtubules. Error bars: SD. Statistical significance was evaluated with a Welch's *t*-test. ****p<0.0001. See *Figure 4—figure supplement 1* for equivalent data when BICD2 was labelled in the Egl/BICD2 complex. (**D** and **E**) Immunoblots of GFP-binding protein pulldowns from *Drosophila* embryo extracts showing RNA-induced co-precipitation of endogenous p150 (**D**) and Dhc (**E**) with Egl::GFP. This effect was observed in four independent experiments. For the blots shown, the amount of extract from which the loaded immunoprecipate was derived was 20 times the amount of extract loaded into the input lane for α-GFP, 200 times the amount of extract loaded into the input lane for α-Dhc and 1000 times the amount of extract loaded into the input lane for α-p150. Thus, only a small fraction of total Egl was associated with p150 and Dhc in the presence of RNA. Embryos expressing free GFP were used as a control. In control experiments, the presence of RNA did not cause co-precipitation of the dynein-dynactin complex with GFP.

DOI: https://doi.org/10.7554/eLife.36312.019

The following source data and figure supplements are available for figure 4:

**Source data 1.** Numerical values for plot in *Figure 4C*.
DOI: https://doi.org/10.7554/eLife.36312.024

**Figure supplement 1.** RNA promotes the association of BICD2 with dynein in the presence of Egl and dynactin.
DOI: https://doi.org/10.7554/eLife.36312.020

**Figure supplement 1—source data 1.** Numerical values for plot in *Figure 4—figure supplement 1C*.
DOI: https://doi.org/10.7554/eLife.36312.021

**Figure supplement 2.** RNA-induced association of Egl/BICD2 with dynein promotes transport.

*Figure 4 continued on next page*

*Figure 4 continued*

DOI: https://doi.org/10.7554/eLife.36312.022

**Figure supplement 2—source data 1.** Numerical values for plots in *Figure 4—figure supplement 2A,B*.

DOI: https://doi.org/10.7554/eLife.36312.023

had a tendency to dissociate upon dilution. This observation suggests dynamic exchange of constituent species. We therefore wondered if the RNA relieves BICD2 autoinhibition by stabilising its interaction with Egl. To test this hypothesis, we first mixed the 59-nt *ILS* RNA with purified Egl/BICD2 and performed size exclusion chromatography. The RNA localisation signal caused a large change in the elution profile of the protein complex compared to the RNA-free form (*Figure 6—figure supplement 1*), indicating a substantial increase in molar mass or a conformational change.

We next used sedimentation equilibrium analytical ultracentrifugation (SE-AUC) to evaluate mean molar masses of complexes in the presence and absence of RNA independently of protein conformation. Over a range of protein concentrations, the presence of the *ILS* caused a large increase in mean molar mass compared to RNA-free samples (*Figure 6A* and *Figure 6—figure supplement 2*).

An orthogonal method for determining molar masses – size-exclusion chromatography with multi-angle light scattering (SEC-MALS) – confirmed that the *ILS* substantially increases the mean molar mass of the Egl/BICD2 sample (*Figure 6B* and *Figure 6—figure supplement 3*). This effect was evident at all salt concentrations examined (*Figure 6B* and *Figure 6—figure supplement 4*). Despite being present in a 10-fold molar access to Egl/BICD2, the scrambled *ILS* RNA elicited a relatively small increase in mean molar mass (*Figure 6C*), confirming selectivity of the Egl/BICD2 complex for an active RNA localisation signal. Our finding that there is some association of Egl/BICD2 with the mutant RNA is compatible with earlier evidence that Egl is not a highly selective RNA-binding protein (*Bullock et al., 2006*; *Dienstbier et al., 2009*; *Dix et al., 2013*). The *ILS* induced a broad range of molar masses in the peak fractions, indicating an equilibrating mixture of larger complexes and smaller constituent components (*Figure 6B,C*). Consistent with such dynamics, the mean molar mass of the peak fractions increased with increasing amounts of Egl/BICD2 and RNA (*Figure 6D*). Analysis of peak SEC-MALS fractions by SDS-PAGE revealed that the *ILS*-induced increases in mass were associated with enhanced interaction of BICD2 and Egl (*Figure 6B* and *Figure 6—figure supplement 4B*). We also used SEC-MALS to determine the effect of the *ILS* on the purified complex of Egl and DmBicD. The mean molar mass of the peak fractions increased substantially in the presence of the *ILS*, and this was again associated with increased binding of Egl and the BicD protein (*Figure 6—figure supplement 5*). Collectively, these experiments reveal that the Egl/BICD2 and Egl/DmBicD complexes readily equilibrate with constituent species and that this is counteracted by the RNA localisation signal.

## The copy numbers of RNA, Egl and BicD in active transport complexes

In our SE-AUC and SEC-MALS experiments, mean molar masses of the mixtures of *ILS*, Egl and a BicD protein could reach ~400 kDa. The predicted molar masses of the BICD2, DmBicD and Egl polypeptides are 93, 89 and 112 kDa, respectively, while the *ILS* has a molar mass of 19 kDa. It was previously shown that DmBicD is a dimer (*Stuurman et al., 1999*), and we confirmed that this is also the case for BICD2 using SEC-MALS (*Figure 6B*; observed molar mass 186.7 ± 0.5 kDa). The mean molar masses observed in our experiments with the *ILS* are therefore compatible with a fraction of BicD dimers being occupied by more than one Egl molecule. To directly evaluate the stoichiometry of Egl and BicD in mRNA transport complexes, we returned to our in vitro motility assay. This system allows investigation of the copy number of these proteins in the fraction of complexes that are able to recruit dynein and dynactin and thus support processive movement on microtubules.

We first produced Egl/BICD2 complexes with SNAP-tagged BICD2 and labelled them with a mixture of SNAP-reactive dyes such that approximately half of BICD2 polypeptides in the preparation were labelled with TMR, and approximately half were labelled with Alexa647. In an idealised situation, the exclusive presence of BICD2 dimers would result in 50% of complexes with one TMR dye and one Alexa647 dye, 25% with two TMR dyes and 25% with two Alexa647 dyes (*Figure 7A*). However, incomplete labelling of SNAP::BICD2 meant that an obligate BICD2 dimer would result in 40% of complexes labelled with both dyes (*Supplementary file 1*). When the labelled Egl/SNAP::BICD2

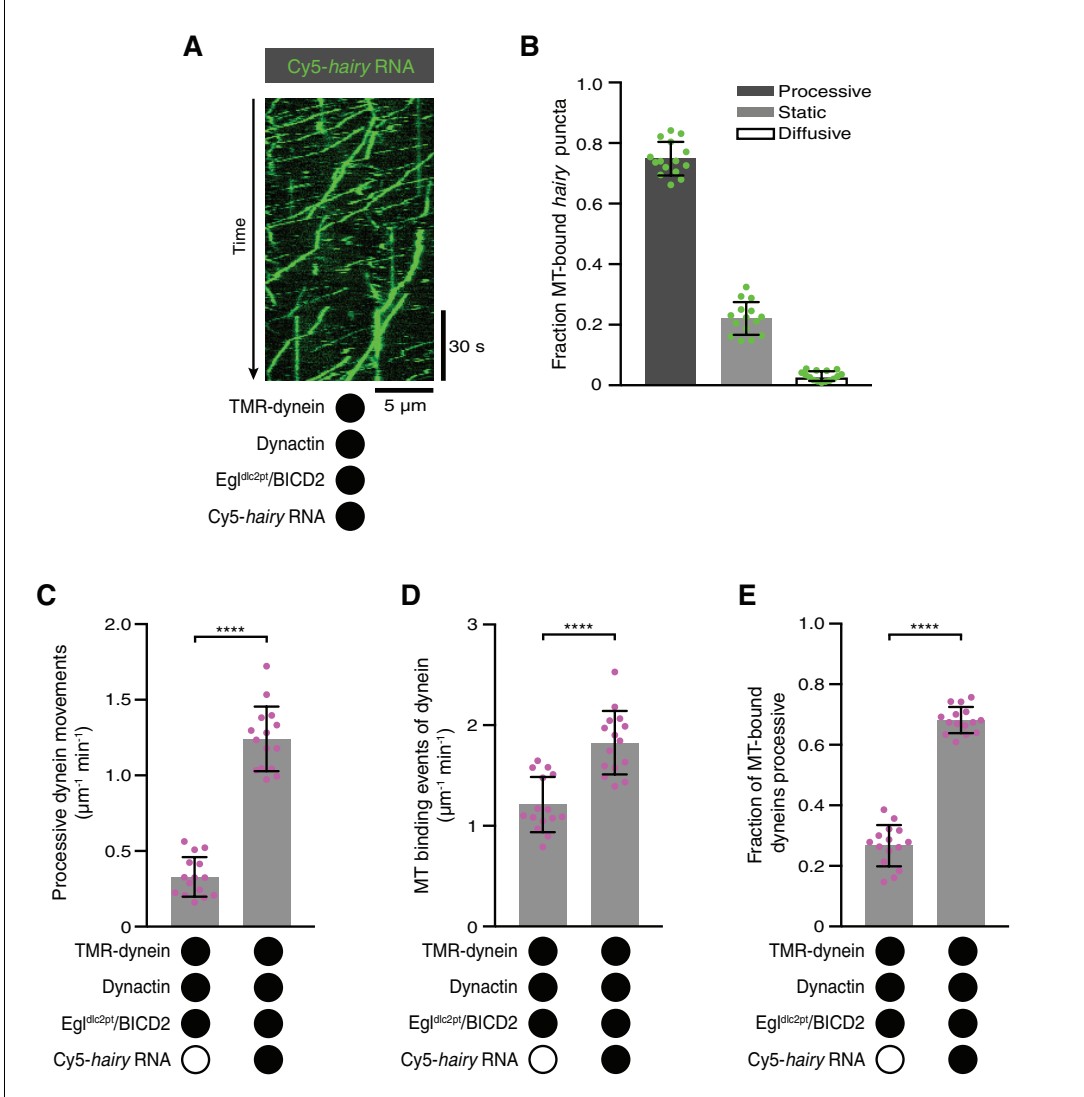

**Figure 5.** The Egl-LC8 interaction is dispensable for RNA-directed activation of dynein motility. (A) Kymograph illustrating robust transport of *hairy* RNA in the presence of dynein, dynactin and the Egl^dlc2pt/BICD2 complex. (B) Fraction of microtubule-associated *hairy* RNA complexes that are processive, static or diffusive using the Egl^dlc2pt/BICD2 complex. (C–E) Numbers of processive dynein movements (C), microtubule-binding events of dynein (D) and fraction of microtubule-binding events that result in processive dynein movements (E) in the presence and absence of *hairy* RNA. In C-E, circles are values for individual microtubules. Error bars: SD. Statistical significance in C-E was evaluated with a Student's *t*-test. ****p<0.0001. See *Figure 5—figure supplement 1* for velocity and run length distributions for these experiments.

DOI: https://doi.org/10.7554/eLife.36312.025

The following source data and figure supplements are available for figure 5:

**Source data 1.** Numerical values for plots in *Figure 5B–E*.
DOI: https://doi.org/10.7554/eLife.36312.028

**Figure supplement 1.** Supplementary data on dynein motility in the presence of dynactin, and Egl^dlc2pt/BICD2 ± *hairy* RNA.
DOI: https://doi.org/10.7554/eLife.36312.026

**Figure supplement 1—source data 1.** Numerical values for plots in *Figure 5—figure supplement 1A,B*.
DOI: https://doi.org/10.7554/eLife.36312.027

sample was used in motility assays with dynein, dynactin and *hairy* RNA, 39% of the motile complexes with a BICD2 signal were labelled with both dyes (*Figure 7B,C*). Our co-localisation analysis therefore fits well with there being a single BICD2 dimer in transport complexes (*Supplementary file 2*).

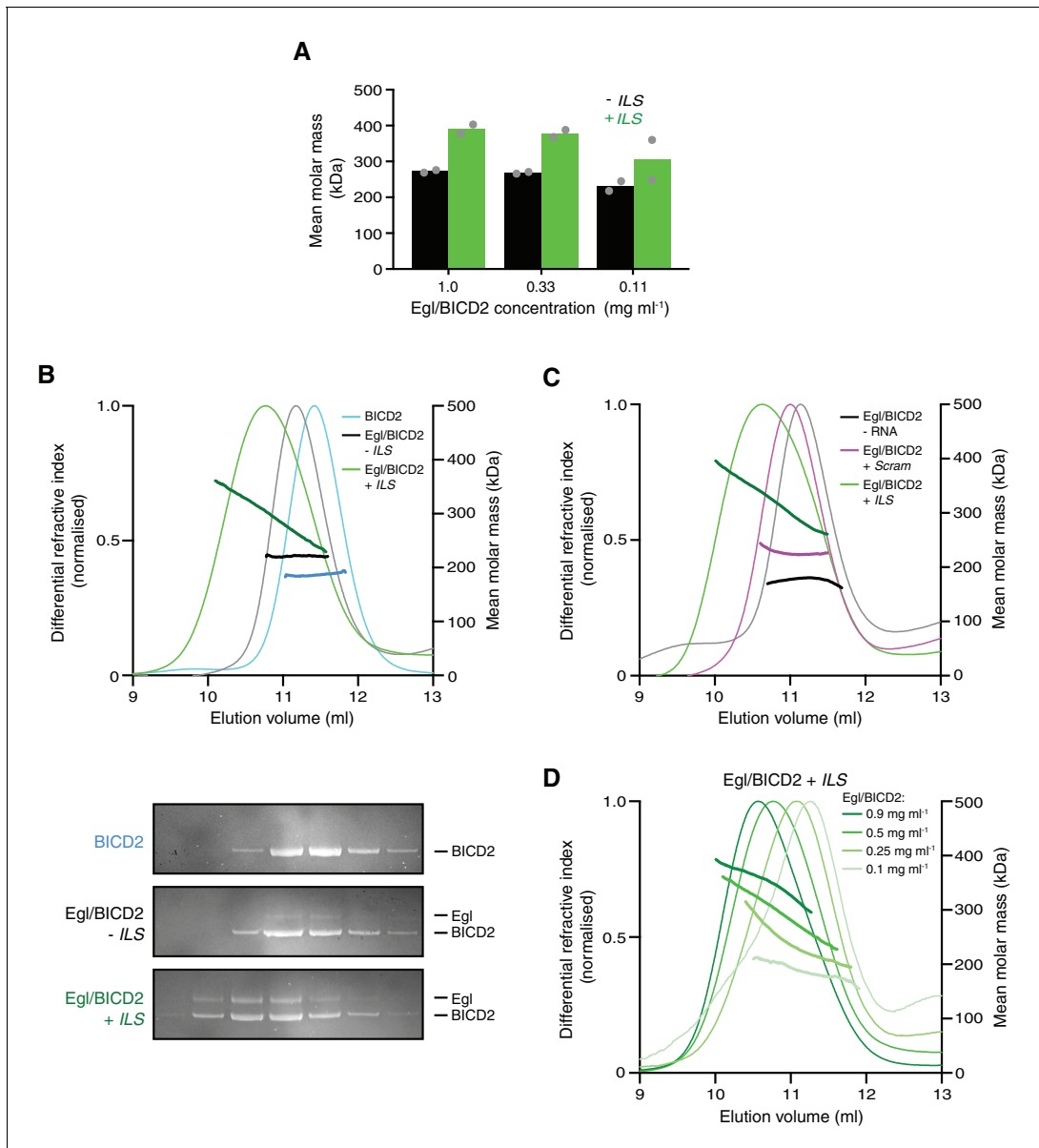

**Figure 6.** The RNA localisation signal promotes the occupancy of BICD2 with Egl. (**A**) Mean molar masses of Egl/BICD2 complexes at different concentrations in the presence and absence of the *ILS* determined by SE-AUC. For comparison, the concentration of Egl/BICD2 in the assembly mix for in vitro motility assays is 0.04 mg ml$^{-1}$. In this and other panels of this figure, the RNA was present in a 10-fold molar excess to the protein (based on an operational assumption of a complex of two Egl molecules and a BICD2 dimer). Circles are values for individual samples. See *Figure 6—figure supplement 2* for examples of raw data and fitting. Experiments were performed in 150 mM salt at 4°C. (**B**) SEC-MALS analysis of samples of Egl/BICD2 in the presence and absence of *ILS* RNA, and BICD2 alone for comparison. The MALS analysis provides the abundance-weighted mean mass of all of the species present throughout the peak (darker lines). Gels of collected fractions stained with SYPRO Ruby reveal more Egl associated with BICD2 in the presence of the *ILS* (maximum Egl:BICD2 ratio without *ILS* = 0.07; maximum Egl:BICD2 ratio with *ILS* = 0.48), which corresponds to species with higher mean molar mass (gels are aligned with corresponding positions in the SEC-MALS trace). Consistent with the relatively modest increase in molar mass compared to BICD2 alone, the SEC-MALS peak for the mixture of Egl/BICD2 without the *ILS* is dominated by free BICD2, with a relatively small amount of Egl. The absence of a BICD2-like shoulder in the trace of this sample presumably reflects rapid binding and unbinding of Egl. Free monomeric Egl elutes later from the column in a broad peak (*Figure 6—figure supplement 3*). The broad range of mean masses across the Egl/BICD2 peak in the presence of *ILS* indicates that our experimental conditions captured an equilibrating mixture of different Egl/BICD2 species. (**C**) SEC-MALS analysis of Egl/BICD2 in the presence of the *ILS*, scrambled *ILS* (*Scram*), or no RNA. In B and C, the concentration of the Egl/BICD2 input was 0.5 mg ml$^{-1}$. (**D**) SEC-MALS analysis of different input concentrations of Egl/BICD2 in the presence of a 10-fold molar excess of the *ILS* (0.5 mg ml$^{-1}$ data are reproduced from B). Note that SEC dilutes proteins ~10 fold before they are subjected to MALS analysis. SEC-MALS experiments were performed in 150 mM salt at room temperature. See *Figure 6—figure supplement 4* for results with Egl/BICD2 ± *ILS* using lower ionic strength buffers.

DOI: https://doi.org/10.7554/eLife.36312.029

*Figure 6 continued on next page*

*Figure 6 continued*

The following source data and figure supplements are available for figure 6:

**Source data 1.** Numerical values for plots in *Figure 6A–D*.
DOI: https://doi.org/10.7554/eLife.36312.040
**Figure supplement 1.** The *ILS* RNA leads to a significant change in the SEC elution profile of the Egl/BICD2 complex.
DOI: https://doi.org/10.7554/eLife.36312.030
**Figure supplement 1—source data 1.** Numerical values for plot in *Figure 6—figure supplement 1*.
DOI: https://doi.org/10.7554/eLife.36312.031
**Figure supplement 2.** Supplementary data for SE-AUC analysis of Egl/BICD2 samples.
DOI: https://doi.org/10.7554/eLife.36312.032
**Figure supplement 2—source data 1.** Numerical values for plots in *Figure 6—figure supplement 2*.
DOI: https://doi.org/10.7554/eLife.36312.033
**Figure supplement 3.** Extended trace from SEC-MALS experiment in *Figure 6B* illustrating typical elution profiles for Egl/BICD2 samples.
DOI: https://doi.org/10.7554/eLife.36312.034
**Figure supplement 3—source data 1.** Numerical values for plot in *Figure 6—figure supplement 3*.
DOI: https://doi.org/10.7554/eLife.36312.035
**Figure supplement 4.** SEC-MALS data for the purified Egl/BICD2 complex in the presence and absence of the *ILS* using buffer with modified salt concentrations.
DOI: https://doi.org/10.7554/eLife.36312.036
**Figure supplement 4—source data 1.** Numerical values for plots in *Figure 6—figure supplement 4A,B*.
DOI: https://doi.org/10.7554/eLife.36312.037
**Figure supplement 5.** SEC-MALS analysis of the purified Egl/DmBicD complex in the presence and absence of *ILS* RNA.
DOI: https://doi.org/10.7554/eLife.36312.038
**Figure supplement 5—source data 1.** Numerical values for plot in *Figure 6—figure supplement 5*.
DOI: https://doi.org/10.7554/eLife.36312.039

When the procedure was repeated with SNAP-tagged Egl co-expressed with BICD2, the proportion of fluorescent complexes that was dual-labelled in the presence of RNA was 37% (*Figure 7D,E*). Correcting for the small fraction of Egl::SNAP molecules that are unlabelled, this result indicates that there are two Egl molecules in the vast majority of active RNA transport complexes (*Supplementary files 1* and *2*). When this experiment was performed in the absence of RNA, the relatively small number of motile Egl complexes observed also had signal from both dyes in 41% of cases (*Figure 7—figure supplement 1*). These data indicate that even when the assembly of the transport machinery is inefficient, motility is usually associated with the presence of two Egl molecules in a complex. The capacity of BicD to bind two Egl molecules is compatible with the symmetrical nature of the Egl-binding region of CC3 (*Liu et al., 2013*).

Finally, we investigated the copy number of RNA in transport complexes by performing motility assays with equimolar amounts of *hairy* RNA preparations that were labelled by random incorporation of either Cyanine3 (Cy3) or Cy5. Of the labelled motile RNPs, 14% had signal from both dyes (*Figure 7F,G*). Considering that the same proportion of complexes should have two Cy3 dyes or two Cy5 dyes, and that the labelling efficiency means that a small fraction of RNA molecules will contain neither dye, these data indicate that 30% of RNPs contained two RNAs and 70% contained one RNA (*Supplementary file 3*). We previously found that transported *hairy* RNPs assembled in *Drosophila* extracts exclusively contain a single RNA (*Amrute-Nayak and Bullock, 2012*; *Soundararajan and Bullock, 2014*). The subset of complexes containing two *hairy* RNAs in our current assay presumably reflects a degree of non-specific RNA-RNA interaction or RNA-protein interaction, which is normally blocked in extracts by the binding of other proteins. We observed very little co-localisation of Cy3-*hairy* and Cy5-*hairy* on a glass surface in the absence of Egl, BICD2 dynein and dynactin (*Figure 7—figure supplement 2*), suggesting that the presence of two copies of the RNA in a subset of transport complexes is predominantly due to an interaction of the second RNA molecule with one of the proteins in the transport machinery. Motile complexes containing both Cy3 and Cy5 exhibited similar velocity distributions and only a modest increase in run length compared to those containing only a single dye (*Figure 7—figure supplement 3*). Thus, the presence of a second RNA does not have substantial functional consequences.

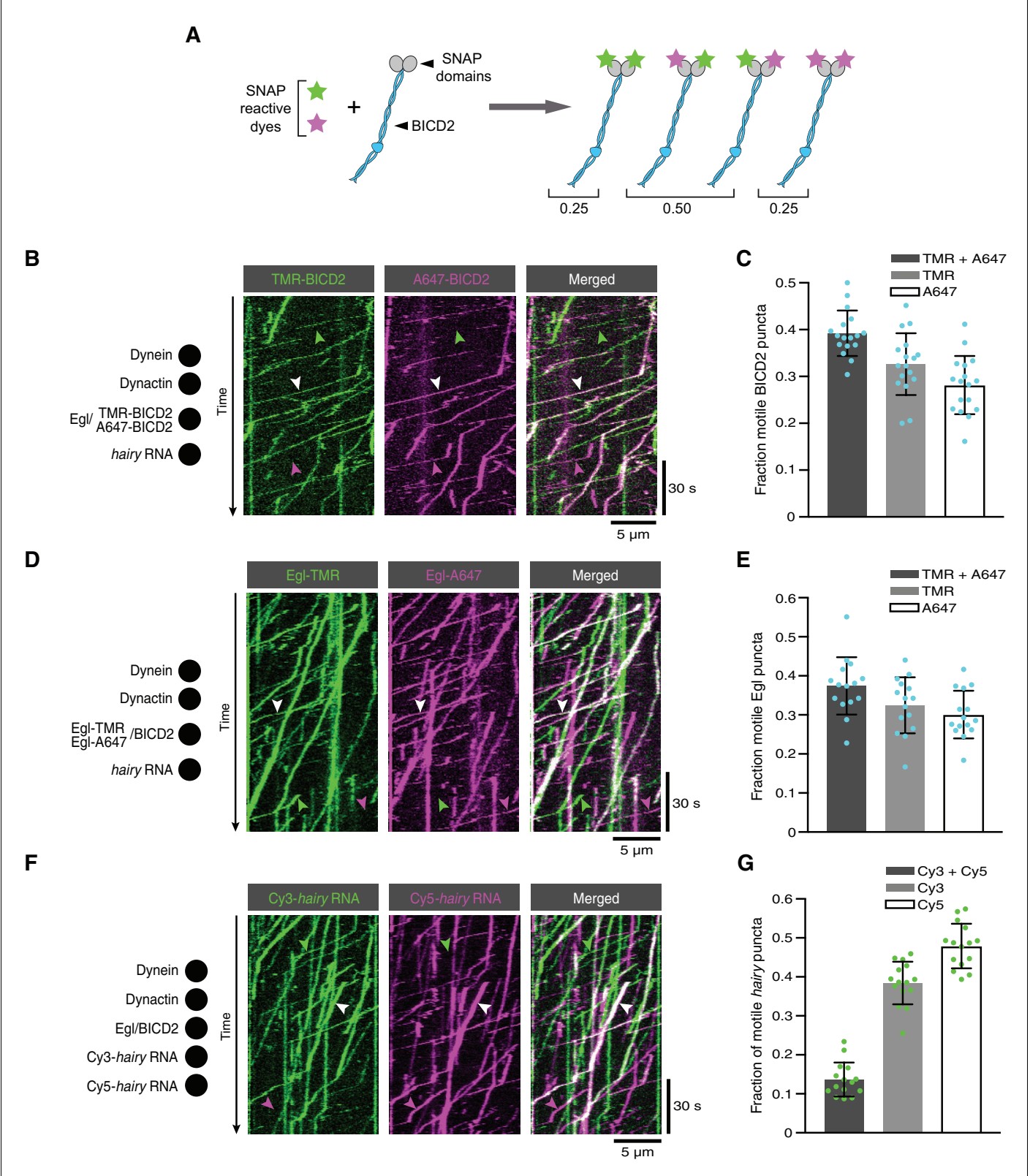

**Figure 7.** The copy number of BICD2, Egl and RNA in active transport complexes. (**A**) Idealised outcome of incubating a SNAP-tagged protein that is present in two copies per complex with equimolar amounts of two different SNAP-reactive dyes. The BICD2 dimer is used as an example, although the same principle applies for experiments with labelled Egl. (**B**) Kymograph of fluorescent signals when a complex of Egl and SNAP::BICD2 is labelled with a mixture of TMR and Alexa647 and assayed in the presence of RNA, dynactin and dynein. (**C**) Fraction of motile BICD2-containing complexes with

*Figure 7 continued on next page*

*Figure 7 continued*

signals from both fluorophores, only TMR, or only Alexa647. (D) Kymograph of fluorescent signals when a complex of Egl::SNAP and BICD2 is labelled with a mixture of TMR and Alexa647 and assayed in the presence of RNA, dynactin and dynein. (E) Fraction of motile Egl-containing complexes labelled with signals from both fluorophores, only TMR, or only Alexa647. (F) Kymograph of fluorescent signals when Cy3-*hairy* and Cy5-*hairy* are mixed and assayed in the presence of Egl/BICD2, dynactin and dynein. (G) Fraction of motile *hairy* RNA puncta labelled with both fluorophores, only Cy3, or only Cy5. In B, D and F, white arrowheads indicate complexes containing both dyes; green and magenta arrowheads indicate, respectively, complexes containing only TMR or only Alexa647 (B and D) or only Cy3 or Cy5 (F). In C, E and G, circles are values for individual microtubules; error bars: SD. See *Supplementary files 1–3* for calculations of copy numbers based on corrections for the proportion of protein or RNA molecules that are unlabelled.

DOI: https://doi.org/10.7554/eLife.36312.041

The following source data and figure supplements are available for figure 7:

**Source data 1.** Numerical values for plots in *Figure 7C,E,G*.

DOI: https://doi.org/10.7554/eLife.36312.048

**Figure supplement 1.** The presence of two Egl proteins in active transport complexes assembled in the absence of RNA.

DOI: https://doi.org/10.7554/eLife.36312.042

**Figure supplement 1—source data 1.** Numerical values for plot in *Figure 7—figure supplement 1B*.

DOI: https://doi.org/10.7554/eLife.36312.043

**Figure supplement 2.** Analysis of RNA-RNA association in the absence of proteins.

DOI: https://doi.org/10.7554/eLife.36312.044

**Figure supplement 2—source data 1.** Numerical values for plot in *Figure 7—figure supplement 2B*.

DOI: https://doi.org/10.7554/eLife.36312.045

**Figure supplement 3.** Assessing the influence of RNA copy number on the motile properties of RNPs.

DOI: https://doi.org/10.7554/eLife.36312.046

**Figure supplement 3—source data 1.** Numerical values for plots in *Figure 7—figure supplement 3A,B*.

DOI: https://doi.org/10.7554/eLife.36312.047

In summary, the results of the dual-labelling experiments are consistent with the vast majority of BicD dimers in active transport complexes associating with two Egl polypeptides, and most associating with a single RNA. Together with our earlier results, these data support a model in which the RNA localisation signal licenses BicD to bind dynein and dynactin by facilitating the association of BicD CC3 with two Egl molecules (see below).

## Discussion

We have succeeded in reconstituting microtubule-based mRNA transport in vitro using purified proteins and have used this system to define a minimal transport-competent RNP. Although genetic experiments indicate that other proteins can modulate the mRNA transport process in vivo (*Dix et al., 2013*; *Hain et al., 2014*), no other factors appear to be obligatory for linkage of the RNA to dynein.

There has recently been considerable focus on the regulation of dynein motility, stemming from the discovery that the isolated N-terminal region of BicD proteins can bridge the interaction of dynein and dynactin and thereby activate transport (*McKenney et al., 2014*; *Schlager et al., 2014*; *Splinter et al., 2012*). Subsequent structural studies have provided important insights into how dynein activity is controlled in this system (*Chowdhury et al., 2015*; *Grotjahn et al., 2018*; *Urnavicius et al., 2018*, *2015*; *Zhang et al., 2017*). However, because full-length BicD proteins interact with dynein and dynactin poorly (*Hoogenraad et al., 2001*, *2003*; *Huynh and Vale, 2017*), it is unclear how dynein activity is controlled within intact cargo-motor complexes.

Previous observations suggested that binding of a cargo-associated protein such as Egl to the C-terminal region of BicD is sufficient to overcome the autoinhibited state of the full-length protein, and thereby lead to recruitment of dynein and dynactin. Our study reveals that a purified Egl/BicD complex does not efficiently associate with dynein and dynactin in the absence of an RNA localisation signal. Thus, the availability of the cargo gates robust activation of dynein motility. This mechanism presumably limits unproductive long-range movement of the motor complex in the absence of an RNA consignment.

We show that the previously reported interaction of Egl with LC8 (*Navarro et al., 2004*) is not required for RNA-directed activation of dynein motility. Characterisation of the features of LC8 that

mediate interaction with its binding partners also argue against a role for LC8 as an adaptor between Egl and dynein; the groove that LC8 uses to associate with the consensus binding motif present in Egl is also used for incorporation into the dynein complex, suggesting mutually exclusive interactions (*Benison et al., 2007*; *Rapali et al., 2011*). However, disrupting the Egl-LC8 interaction in *Drosophila* does significantly compromise the function of Egl in the maintenance of oocyte fate (*Navarro et al., 2004*). There are several examples of LC8 acting as a chaperone for binding partners independently of its association with dynein (*Rapali et al., 2011*), and it may serve the same function for Egl in vivo.

Our data indicate that a key consequence of RNA binding to Egl is stimulation of the interaction of BicD with dynein and dynactin. Thus, RNA-bound Egl must overcome the autoinhibition of full-length BicD that prevents CC1/2 from engaging with dynein and dynactin. Negative stain electron microscopy in a contemporary study (*Sladewski et al., 2018*) lends further support to this notion; a folded back conformation of full-length DmBicD (*Stuurman et al., 1999*), which is likely to represent the autoinhibited state (*Hoogenraad et al., 2001*, *2003*; *Liu et al., 2013*), was retained in the presence of Egl alone, but not detected in the presence of both RNA and Egl.

Our single-molecule analysis is also consistent with the activation of transport by RNA being mediated by CC1/2. The recruitment of RNA to dynein by Egl/BICD2 is dependent on dynactin (*Figure 2B,C*), as is also the case for the interaction of BICD2N with the motor complex (*McKenney et al., 2014*). Furthermore, the ability of RNA-bound Egl/BICD2 to augment dynein's binding to microtubules (*Figure 3C*), as well as its velocity (*Figure 3E*) and run length (*Figure 3F*), is also shared with BICD2N (*McKenney et al., 2014*). Very recently, it has been shown that a single BICD2N dimer and a single dynactin can recruit one or two dynein complexes (*Grotjahn et al., 2018*; *Urnavicius et al., 2018*), with the binding of the second motor increasing velocity and run length (*Urnavicius et al., 2018*). The two-motor state is associated with a subtle difference in the position of the N-terminal region of BICD2 CC1 (*Urnavicius et al., 2018*). *Sladewski et al., 2018* provide evidence that two dynein motors are present in the majority of their transport RNPs, raising the possibility that interaction of RNA-bound Egl with BICD2 modulates the velocity and run length of transport complexes by favouring the two-motor-binding conformation of CC1.

An in vitro study of a yeast actin-based mRNA transport complex also reported stimulation of processive movement by the RNA cargo (*Sladewski et al., 2013*) (although an independent investigation of the same complex reported no influence of the RNA [*Heym et al., 2013*]). The mechanism that we and *Sladewski et al., 2018* propose for RNA-mediated activation of dynein – involving relief of autoinhibition of an adaptor – is distinct from the one proposed for the yeast transport complex, which is based on RNA-dependent dimerisation of monomers of the myosin motor (*Sladewski et al., 2013*). Thus, multiple strategies may have evolved to co-ordinate the processivity of cytoskeletal motors with the availability of an RNA cargo.

How could binding of RNA to Egl relieve BicD autoinhibition? Although a complex of Egl bound to BicD can be purified in the absence of RNA following overexpression in insect cells, our data indicate that it readily dissociates into constituent species. The interaction between Egl and BicD is mediated by the first 79 amino acids of Egl and a 42-amino-acid region of BicD CC3 (*Dienstbier et al., 2009*; *Liu et al., 2013*). Although we cannot rule out additional mechanisms of RNA-mediated activation of BicD, the most parsimonious explanation for our data is that the RNA localisation signal promotes the occupancy of CC3 with Egl, thereby freeing CC1/2 to interact with dynein and dynactin (*Figure 8*). It is not clear how binding of RNA-associated Egl (or Rab6$^{GTP}$ for that matter) to BicD CC3 releases CC1/2. One possibility is that binding of Egl/Rab6$^{GTP}$ competes directly with the interaction of CC3 with the N-terminal sequences. Alternatively, occupancy of the Rab6$^{GTP}$- and Egl-binding site in CC3 could induce changes in coiled-coil architecture that are propagated along the molecule to release a discrete autoinhibitory interaction (*Liu et al., 2013*). The discovery of crystal forms of CC3 with different coiled-coil registers (*Liu et al., 2013*; *Terawaki et al., 2015*) lends support to the involvement of coiled-coil dynamics in the activation of BicD.

It is striking that the vast majority of active transport complexes contain two Egl polypeptides per BicD dimer, even when the transport process is compromised by the omission of RNA. This finding suggests that occupancy of both Egl-binding sites of BicD CC3 favours the relief of BicD autoinhibition and recruitment of dynein and dynactin. How could the RNA localisation signal stabilise the heterotetrameric Egl/BicD complex? Purified BicD was found not to interact directly with RNA localisation signals (*Dienstbier et al., 2009*), suggesting that the RNA does not act as a bridge

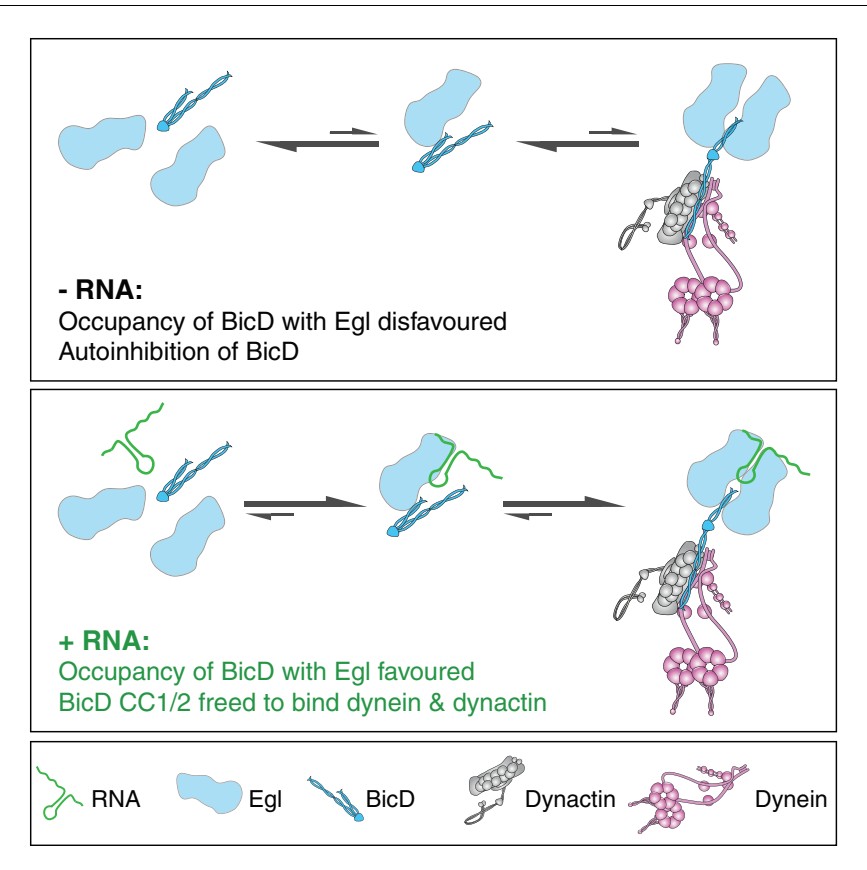

**Figure 8.** Model for the mechanism of RNA-stimulated assembly of an active dynein-dynactin complex. The RNA favours the interaction of Egl with CC3 of BicD, which promotes release of CC1/2 of BicD to interact with dynein and dynactin. A single RNA molecule is shown in the transport complex as our data indicate that this scenario is common.

DOI: https://doi.org/10.7554/eLife.36312.049

between Egl and BicD. Our single-molecule experiments are consistent with two Egl proteins being able to associate with a single RNA molecule. A structure-function study of an Egl-binding RNA localisation signal revealed two structurally-related helices that must be precisely registered with each other in order to trigger mRNA transport (*Bullock et al., 2010*). Our SEC-MALS analysis indicates that free Egl is monomeric (*Figure 6—figure supplement 3*). It is therefore tempting to speculate that the two helices of the RNA localisation signal are discrete binding sites for Egl monomers, as this offers a simple explanation for how the RNA facilitates the association of two Egl molecules with a BicD dimer. Alternatively, binding of the RNA localisation signal could induce a conformational change in Egl that stabilises the protein or increases its affinity for CC3, thereby favouring full occupancy of BicD. High-resolution structures of RNA-protein complexes will be required to discriminate between these possibilities.

In addition to $Rab6^{GTP}$-associated vesicles and Egl-associated mRNAs, BicD proteins are implicated in transport of a diverse range of cellular cargoes and pathogens by dynein and dynactin (*Dharan et al., 2017*; *Hoogenraad and Akhmanova, 2016*; *Indran et al., 2010*; *Redwine et al., 2017*). It is conceivable that the cargo also promotes the activation of dynein in these systems by scaffolding the association of two cargo-associated proteins with CC3. It is easy to envisage how this could occur during membrane trafficking, when the diffusion of CC3-interacting proteins within the membrane would greatly facilitate co-incident association. The finding that BicD CC3 can simultaneously bind two $Rab6^{GTP}$ proteins (*Liu et al., 2013*) is compatible with this scenario.

# Materials and methods

## Key resources table

| Reagent type (species) or resource | Designation | Source or reference | Identifiers | Additional information |
|---|---|---|---|---|
| Recombinant DNA reagent (*Drosophila melanogaster*) | Egalitarian (Egl) cDNA | Epoch Life Sciences | Corresponding to NCBI:NM_166623 | Codon optimised for Sf9 cell expression |
| Recombinant DNA reagent (*D. melanogaster*) | Bicaudal-D (BicD) cDNA | Epoch Life Sciences | Corresponding to NCBI:NM_165220 | Codon optimised for Sf9 cell expression |
| Recombinant DNA reagent (*Mus musculus*) | Bicaudal-D2 (BICD2) cDNA | Epoch Life Sciences | Corresponding to NCBI:NM_001039179 | Codon optimised for Sf9 cell expression |
| Recombinant DNA reagent (*Homo sapiens*) | Dynein heavy chain (DHC) cDNA | Epoch Life Sciences; PMID:24986880 | Corresponding to NCBI:NM_001376.4 | Codon optimised for Sf9 cell expression |
| Recombinant DNA reagent (*H. sapiens*) | Dynein intermediate chain 2 (DIC2) cDNA | Epoch Life Sciences; PMID:24986880 | Corresponding to NCBI:AF134477 | Codon optimised for Sf9 cell expression |
| Recombinant DNA reagent (*H. sapiens*) | Dynein light intermediate chain 2 (DLIC2) cDNA | Epoch Life Sciences; PMID:24986880 | Corresponding to NCBI:NM_006141.2 | Codon optimised for Sf9 cell expression |
| Recombinant DNA reagent (*H. sapiens*) | Dynein light chain Tctex (Tctex) cDNA | Epoch Life Sciences; PMID:24986880 | Corresponding to NCBI:NM_006519.2 | Codon optimised for Sf9 cell expression |
| Recombinant DNA reagent (*H. sapiens*) | Dynein light chain LC8 (LC8) cDNA | Epoch Life Sciences; PMID:24986880 | Corresponding to NCBI:NM_003746.2 | Codon optimised for Sf9 cell expression |
| Recombinant DNA reagent (*H. sapiens*) | Dynein light chain Roadblock (Robl) cDNA | Epoch Life Sciences; PMID:24986880 | Corresponding to NCBI:NM_141183.3 | Codon optimised for Sf9 cell expression |
| Recombinant DNA reagent | pAceBac1 plasmid | PMID:27165327 | | |
| Recombinant DNA reagent | pIDC plasmid | PMID:27165327 | | |
| Recombinant DNA reagent (*D. melanogaster*) | *hairy* 3'UTR plasmid | PMID:12743042 | | |
| Recombinant DNA reagent (*D. melanogaster*) | *I-factor* plasmid | PMID:15992540 | | |
| Sequence-based reagent | *ILS* RNA 5'.AAUGCACACCUCCCUCGUCACUCUUGAUUUUUCAAGAGCCUUCGAUCGAGUAGGUGUGCA.3' | GE Dharmacon | | With or without 5' Dy647 label |
| Sequence-based reagent | *ILS* scram RNA 5'.AAAAUGUGGUGCACUAUCUUCGUAUUCCAGUGCCACCGUGGUCUAAUUCACUCGUCGCC.3' | GE Dharmacon | | With or without 5' Dy547 label |
| Cell line (*Spodoptera frugiperda*) | Sf9 | ThermoFisher Scientific | ThermoFisher Scientific: 11496015 | Mycoplasma-free |
| Genetic reagent (*D. melanogaster*) | *P[tub-Egl::GFP]* | PMID:19515976 | FLYB:FBal0230300 | |
| Genetic reagent (*D. melanogaster*) | *Sco/CyO P[actin5C-GFP]* | Bloomington *Drosophila* Stock Center | FLYB: FBst0004533; RRID:BDSC_4533 | |
| Antibody | anti-GFP (mouse monoclonal) | Sigma Aldrich | Sigma-Aldrich:11814460001; RRID:AB_390913 | Mix of clones 7.1 and 13.1 (1:1000) |
| Antibody | anti-*D. melanogaster* Dhc (mouse monoclonal) | Developmental Studies Hybridoma Bank; PMID:10637305 | DSHB:2C11-2; RRID:AB_2091523 | (1:1000) |
| Antibody | anti-*D. melanogaster* p150-C-term (rabbit polyclonal) | PMID:17325206 | | Raised against aa 1,073–1,280 (1:10,000) |
| Commercial assay, kit | GFP-trap magnetic agarose beads | Chromotek | Chromotek:gtma-20 | |
| Commercial assay, kit | Coomassie protein assay kit | ThermoFisher Scientific | ThermoFisher Scientific: 23200 | |

*Continued on next page*

*Continued*

| Reagent type (species) or resource | Designation | Source or reference | Identifiers | Additional information |
|---|---|---|---|---|
| Commercial assay, kit | Full-Range Rainbow prestained molecular weight markers | GE Healthcare | GE Healthcare:RPN800E | |
| Commercial assay, kit | Coomassie Instant Blue protein stain | Expedeon | Expedeon:ISB1L | |
| Commercial assay, kit | MEGAScript T7 transcription kit | ThermoFisher Scientific | ThermoFisher Scientific: AM1333 | |
| Commercial assay, kit | MEGAScript SP6 transcription kit | ThermoFisher Scientific | ThermoFisher Scientific: AM1330 | |
| Chemical compound, drug | Alexa488-UTP | ThermoFisher Scientific | ThermoFisherScientific: C11403 | |
| Chemical compound, drug | Cy3-UTP | PerkinElmer | PerkinElmer:NEL582001EA | |
| Chemical compound, drug | Cy5-UTP | PerkinElmer | PerkinElmer: NEL583001EA | |
| Chemical compound, drug | SNAP-Cell TMR-Star | New England Biolabs | NEB:S9105S | |
| Chemical compound, drug | SNAP-Surface Alexa Fluor 647 | New England Biolabs | NEB:S9136S | |
| Chemical compound, drug | PEG | Rapp Polymere | Rapp Polymere:103000–20 | |
| Chemical compound, drug | Biotin-PEG | Rapp Polymere | Rapp Polymere: 133000-25-20 | |
| Chemical compound, drug | PLL-g-PEG | Susos AG | Susos AG:PLL(20)-g[3.5]-PEG(2) | |
| Chemical compound, drug | Pluronic-F127 | Sigma-Aldrich | Sigma-Aldrich:P2243 | |
| Chemical compound, drug | Paclitaxel (taxol) | Sigma-Aldrich | Sigma-Aldrich:T1912 | |
| Chemical compound, drug | GMPCPP | Jena Bioscience | Jena Bioscience:NU-405 | |
| Other, native protein | Glucose oxidase | Sigma-Aldrich | Sigma-Aldrich:G2133 | |
| Other, native protein | Catalase | Sigma-Aldrich | Sigma-Aldrich:C40 | |
| Other, native protein | Streptavidin | Sigma-Aldrich | Sigma-Aldrich:S4762 | |
| Other, native protein | α-casein | Sigma-Aldrich | Sigma-Aldrich:C6780 | |
| Other, native protein | Porcine tubulin, unlabelled | Cytoskeleton Inc. | Cytoskeleton Inc:T240 | |
| Other, native protein | Porcine tubulin, biotin-conjugated | Cytoskeleton Inc. | Cytoskeleton Inc:T333P | |
| Other, native protein | Porcine tubulin, HiLyte 488-conjugated | Cytoskeleton Inc. | Cytoskeleton Inc:TL488M | |
| Software, algorithm | FIJI | PMID:22743772 | RRID:SCR_002285 | |
| Software, algorithm | Prism | Graphpad | RRID:SCR_002798 | |
| Software, algorithm | Sednterp | T. Laue (University of New Hampshire) | RRID:SCR_016253 | |
| Software, algorithm | SEDPHAT 13b | PMID:12895474 | RRID:SCR_016254 | |
| Software, algorithm | GUSSI | PMID:26412649 | RRID:SCR_014962 | |
| Software, algorithm | ASTRA | Wyatt | RRID:SCR_016255 | |

## Cell lines

Sf9 cells (ThermoFisher Scientific, Waltham, MA) have not been genetically profiled since purchase but were grown in a tissue culture facility dedicated to insect cell expression. The cells were tested for mycoplasma twice a year (MycoAlert Detection Kit, Lonza) and the results were always negative.

## Cloning and recombinant protein expression

Sequences encoding Egalitarian and BicD proteins (*Drosophila melanogaster* Egl isoform B: NM_166623, mouse BICD2:NM_001039179 and *Drosophila melanogaster* BicD:NM_165220) were synthesised commercially (Epoch Life Sciences, Sugar Land, TX) with codons optimised for expression in *Spodoptera frugiperda* Sf9 cells, and cloned for use with the MultiBac expression system. Where required, sequences encoding $SNAP_f$ tags for fluorescent labelling of protein complexes and ZZ-LTLT tags for IgG-based affinity purification (*Reck-Peterson et al., 2006*) were added by Gibson Assembly (NEB, Ipswich, MA) of PCR-amplified insert and backbone fragments. All constructs were validated by sequencing of the entire open-reading frame. Genes encoding Egl::LTLT-ZZ or Egl:: SNAP-LTLT-ZZ were cloned downstream of the *polh* promoter of the pACEBac1 acceptor vector (*Sari et al., 2016*), while genes encoding BICD2, SNAP::BICD2, or *Drosophila melanogaster* BicD (DmBicD) were cloned downstream of the *polh* promoter of the pIDC donor vector (*Sari et al., 2016*). The donor and acceptor vectors were recombined at defined Cre loci and incorporated into the baculovirus genome for simultaneous co-expression of Egl and BicD proteins. The same strategy was used for assembly of the gene encoding human DHC (tagged at the N-terminus with ZZ-LTLT-SNAP) with those encoding other human dynein subunits, as described previously (*Schlager et al., 2014*). The isoform composition of the assembled dynein complex is as follows: DHC:NM_001376.4; DIC2:AF134477; DLIC2:NM_006141.2; Tctex:NM_006519.2; LC8:NM_003746.2 and Robl: NM_014183.3. All recombinant proteins were expressed from the baculovirus genome in Sf9 cells as described previously (*Schlager et al., 2014*). Following protein expression, cells were frozen in liquid $N_2$ and stored at $-80°C$.

## Site-directed mutagenesis

The $Egl^{dlc2pt}$ mutations (S965K + S969R) (*Navarro et al., 2004*) were generated by whole-vector PCR using a single pair of complementary mutagenic primers containing the desired sequence. Following amplification, the template DNA was digested with DpnI, and the amplicon ligated and propagated by transformation into α-Select Silver Efficiency chemically competent *E. coli* (Bioline, London, UK). The presence of the desired mutations, and no others, was confirmed by sequencing of the entire open-reading frame.

## Protein purification

All purification steps were performed at 4°C. Native dynactin was purified from pig brain as described previously (*Schlager et al., 2014*; *Urnavicius et al., 2015*). Dynein, BICD2, Egl/BICD2 and Egl/DmBicD complexes were affinity purified via an N-terminal ZZ-LTLT on DHC (ZZ-LTLT-SNAP:: DHC) and BICD2 (ZZ-LTLT-BICD2), or a C-terminal LTLT-ZZ tag on Egl (Egl::LTLT-ZZ or Egl::SNAP-LTLT-ZZ). Frozen Sf9 cells were thawed on ice. For dynein purification, cells were resuspended in lysis buffer (50 mM HEPES pH 7.3, 100 mM NaCl, 10% glycerol, 1 mM DTT, 0.1 mM MgATP, 2 mM PMSF, 1 x cOmplete EDTA-free protease inhibitor cocktail (Sigma-Aldrich, St Louis, MO)). For purification of Egl/BICD2 and Egl/DmBicD complexes, lysis buffer was modified to include 500 mM NaCl to disrupt any association of Egl with native RNA species. Lysates were generated by repeated passage of resuspended cells through a Wheaton dounce tissue grinder (Fisher Scientific, Hampton, NH) and subsequently clarified by ultracentrifugation at 70,000 RPM (504,000 x *g*) using a Beckman Coulter Type 70 Ti fixed-angle rotor in a Beckman Coulter Optima L-100 XP preparative ultracentrifuge.

During centrifugation, IgG Sepharose 6 affinity resin (GE Healthcare Life Sciences, Little Chalfont, UK) was applied to a gravity flow Econo-column (Bio-Rad, Hercules, CA) and washed twice with five column volumes of lysis buffer (typically 5 ml of resin slurry was used). Clarified lysate was added directly to the affinity matrix in the column, which was then sealed and agitated by gentle rolling for 3 hr. After incubation, the lysate was allowed to flow through the column by gravity and the retained affinity matrix washed twice with five column volumes of lysis buffer and twice with five column

volumes of TEV buffer (50 mM Tris-HCl pH 7.4, 150 mM KOAc, 2 mM MgOAc, 1 mM EGTA-KOH pH 7.3, 10% glycerol). If required, bound SNAP-tagged proteins were fluorescently labelled on-column before proceeding to elution (see below). Bound proteins were eluted by overnight TEV cleavage of the ZZ affinity tag using gentle rolling agitation and ~0.03 mg ml$^{-1}$ TEV protease in a final volume of 15 ml TEV buffer. Eluted protein was recovered by gravity flow through a fresh Econo-column and concentrated to ~1.5 mg ml$^{-1}$ with a 100 kDa MWCO Amicon Ultra-4 centrifugal filter unit (Merck, Darmstadt, Germany).

The affinity-purified protein complexes were further purified by FPLC-based gel-filtration chromatography (AKTA Purifier and AKTA Micro, GE Healthcare Life Sciences) in GF150 buffer (25 mM HEPES pH 7.3, 150 mM KCl, 1 mM MgCl$_2$, 5 mM DTT, 0.1 mM MgATP, 10% glycerol) to remove large aggregates, TEV protease, and other small contaminants. For SEC-MALS and SE-AUC experiments, GF150 was modified to include 5 mM TCEP instead of DTT. For the dynein complex, a TSKgel G4000SWxl with guard column (TOSOH Bioscience Ltd, Reading, UK) was used, while a Superose 6 Increase 3.2/300 column (GE Healthcare Life Sciences) was used for BICD2, Egl/BICD2 and Egl/DmBicD complexes. Fractions containing the dynein complex were pooled and concentrated to ~1 mg ml$^{-1}$. Fractions containing BICD2, Egl/BICD2 or Egl/DmBicD complexes were pooled without an additional concentration step. All purified proteins were dispensed in aliquots for single use, flash frozen in liquid N$_2$, and stored at −80°C. Protein concentrations were determined using a Coomassie Protein Assay Kit (ThermoFisher Scientific). To assess purity, proteins were resolved by SDS-PAGE using Novex 4–12% Bis-Tris precast gels (ThermoFisher Scientific) and MES-SDS running buffer. Protein bands were visualised using Coomassie Instant Blue protein stain (Expedeon, Over, UK) and imaged with a ChemiDoc XRS + system (Bio-Rad). Protein sizes were evaluated by comparison with Full-Range Rainbow prestained molecular weight markers (GE Healthcare Life Sciences).

## Fluorescent labelling of SNAP-tagged proteins

Fluorescent labelling of SNAP-tagged proteins with either SNAP-Cell TMR-Star (NEB) or SNAP-Surface Alexa Fluor 647 (NEB) was performed on-column during affinity capture according to a previously described method that labels >95% of dynein dimers with at least one dye (*Schlager et al., 2014*). For the mixed-labelling of SNAP::BICD2 and Egl::SNAP in *Figure 7*, an extended labelling time of 4 hr and a further 10-fold excess of total SNAP-fluorophore reagent was used. This method labelled 90% of SNAP-tagged polypeptides (81% of complexes containing two protein copies labelled with two dyes) (*Supplementary file 1*). Labelling efficiency was determined with spectrophotometry as previously described (*Schlager et al., 2014*). The ratio of SNAP-Surface Alexa Fluor 647 to SNAP-Cell TMR-Star that yielded approximately half of labelled polypeptides having one fluorophore and half the other fluorophore was determined empirically for different batches of the dyes.

## RNA synthesis and purification

Uncapped Cy5-*hairy* RNA or Cy3-*hairy* RNA was transcribed in vitro from a gel-purified PCR amplicon template using the MEGAscript T7 Transcription Kit (Ambion). The RNA is a 730-nt region of the 3'UTR containing the RNA localisation signal (*Bullock et al., 2003*). Cy3-UTP or Cy5-UTP (PerkinElmer, Waltham, MA) was added to the transcription reaction together with a 4-fold excess of unlabelled UTP in order to label the RNA at multiple internal sites. Alexa488-*hairy* RNA was synthesised from the same template using a 1:9 ratio of Alexa488-UTP (ThermoFisher Scientific) to unlabelled UTP. Cy5-*I-factor* RNA was synthesised from a linearised plasmid template using the MEGAscript SP6 Transcription Kit (Ambion) and a 1:3 ratio of Cy5-UTP to unlabelled UTP. The RNA is 597-nt long and contains the *ILS* localisation signal (*Van De Bor et al., 2005*). Following digestion of the template DNA with DNase I, proteins were removed using phenol-chloroform-isoamyl alcohol (ThermoFisher Scientific). Synthesised RNA was separated from unincorporated nucleotides by two rounds of purification with Sephadex G-50 size-exclusion RNA spin columns (Sigma-Aldrich), precipitated with NH$_4$OAc/ethanol and resuspended in nuclease-free dH$_2$O. These procedures typically yield RNA samples with an average of ~3 dyes per molecule. Where relevant, the mean number of dyes per RNA molecule was determined by spectrophotometry (*Supplementary file 3*). *ILS* wild-type (*Van De Bor et al., 2005*) and scrambled mutant RNAs (with and without a single 5' DY547 or

DY647 dye) were synthesised, decapped, deprotected, and HPLC purified by GE Dharmacon (Lafayette, CO). An additional two A's were included at the 5' prime of synthetic RNAs to space the fluorophore from the wild-type or mutant localisation signal. Sequences of the RNAs can be found in the Key Resources Table. For SE-AUC and SEC-MALS experiments, RNAs were further purified by gel-filtration chromatography in GF150 buffer (Superose 6 Increase 3.2/300, AKTA Micro (GE Healthcare)). All RNA concentrations were determined by spectrophotometry.

## Motility chamber preparation

Glass surfaces were prepared as described previously (Bieling et al., 2010). Motility chambers with a volume of ~10 µl were assembled by adhering glass cover slips functionalised with PEG/Biotin-PEG (Rapp Polymere, Tuebingen, Germany) to glass slides passivated with PLL-g-PEG (SuSos AG, Duebendorf, Switzerland) using three segments of double-sided tape distributed along the width of the slide. The arrangement of tape yielded two parallel motility chambers per cover slip and allowed side-by-side comparison of two different conditions on the same glass surface. For the experiment presented in *Figure 7—figure supplement 2*, RNA samples were added to the imaging chambers at this point. For all other assays, chamber surfaces were further passivated for 5 min with 1% (w/v) Pluronic F-127 (Sigma-Aldrich) and washed twice with 20 µl chilled motility buffer (30 mM HEPES pH 7.3, 5 mM $MgSO_4$, 1 mM EGTA pH 7.3, 1 mM DTT, 0.5 mg ml$^{-1}$ BSA). Chambers were then incubated with 2 mg ml$^{-1}$ streptavidin (Sigma-Aldrich) for 5 min and again washed twice with 20 µl motility buffer. To block any unpassivated surface, chambers were incubated with 20 mg ml$^{-1}$ α-casein (Sigma-Aldrich) for 5 min and washed twice with 20 µl motility buffer. The prepared chambers were kept in a humidified container until the addition of microtubules and protein/RNA mixtures to prevent desiccation of chamber surfaces.

## Polymerisation and stabilisation of microtubules

Microtubules were polymerised from porcine tubulin (Cytoskeleton Inc., Denver, CO) and labelled with fluorophores and biotin by stochastic incorporation of labelled dimers into the microtubule lattice. Mixes of 1.66 µM unlabelled tubulin, 0.15 µM Hilyte488-tubulin, and 0.4 µM biotin-tubulin were incubated in BRB80 (80 mM PIPES pH 6.85, 2 mM $MgCl_2$, 0.5 mM EGTA, 1 mM DTT) with 0.5 mM GMPCPP (Jena Bioscience, Jena, Germany) for 2–4 hr at 37°C. Polymerised microtubules were pelleted in a room temperature table top centrifuge at 18,400 x *g* for 8.5 min, and washed once with pre-warmed (37°C) BRB80. After pelleting once more, the microtubules were gently resuspended in pre-warmed (37°C) BRB80 containing 40 µM paclitaxel (taxol; Sigma-Aldrich) and used on the same day.

## In vitro motility assay

Constituents of motility assays were incubated together on ice for 1–2 hr by dilution into motility buffer to the following concentrations: 100 nM dynein, 200 nM dynactin, 100 nM Egl/BICD2 or Egl/DmBicD (using the operational assumption of two Egl molecules and one dimer of the BicD protein per complex), and 1 µM RNA. To ensure that all complexes assemble at the same ionic strengths, KCl was supplemented to a final concentration of 50 mM during assembly. Just prior to imaging, stabilised microtubules were immobilised in a prepared motility chamber for 5 min and subsequently washed once with motility buffer that also contained 50 mM KCl, 1 mg ml$^{-1}$ α-casein, and 20 µM taxol. Assembly mixes were then diluted 40-fold (with the exception of the complexes in *Figure 4A–C*, which were diluted 20-fold) in motility buffer that also contained 50 mM KCl, 1 mg ml$^{-1}$ α-casein, 20 µM taxol, 2.5 mM MgATP, and an oxygen scavenging system (1.25 µM glucose oxidase, 140 nM catalase, 71 mM 2-mercaptoethanol, 25 mM glucose) that greatly limits photobleaching (*Yildiz et al., 2003*). Diluted assembly mixes were applied to immobilised microtubules in the motility chamber for imaging at room temperature (23 ± 1°C). For the experiment documented in *Figure 7—figure supplement 2*, RNA only was added to the chamber and immediately washed with motility buffer containing 50 mM KCl, 1 mg ml$^{-1}$ α-casein, 20 µM taxol, 2.5 mM MgATP, and an oxygen scavenging system.

## TIRF microscopy

For each chamber, a single multicolour acquisition of 500 frames was made at the maximum achievable frame rate (~2 frames s$^{-1}$) and 100 ms exposure per frame using a Nikon TIRF microscope system controlled with Micro-Manager open-source acquisition software (*Edelstein et al., 2010*) and equipped with a Nikon 100 × oil objective (APO TIRF, 1.49 NA oil). For the experiment documented in *Figure 7—figure supplement 2*, single frames were captured with a 1 s exposure in each channel. The following lasers were used: Coherent Sapphire 488 nm (150 mW), Coherent Sapphire 561 nm (150 mW), Coherent CUBE 641 nm (100 mW). Images were captured with an iXon$^{EM}$+ DU-897E EMCCD camera (Andor, Belfast, UK), resulting in pixel dimensions of 105 x 105 nm. Multicolour acquisitions used sequential image capture with switching of emission filters (GFP, Cy3, and Cy5 (Chroma Technology Corp., Bellows Falls, VT)).

## Immunoprecipitation from *Drosophila* extracts

Extracts were generated from embryos of *P[tub-Egl::GFP]* (*Dienstbier et al., 2009*) or *Sco/CyO P [actin5C-GFP]* flies (Bloomington *Drosophila* Stock Center: RRID:BDSC_4533), which contain genomically-integrated transgenes expressing Egl::GFP or GFP from the ubiquitous $\alpha$-tubulin or $\beta$-actin promoters, respectively. 0–12 hr embryos were dechorionated and flash frozen in liquid $N_2$. 300 µl chilled extraction buffer (25 mM HEPES pH 7.3, 50 mM KCl, 1 mM MgCl$_2$, 2 mM DTT, 2x cOmplete EDTA-free protease inhibitor) was added for each 100 mg of frozen embryos, followed by grinding on ice with a motorised pellet pestle (ThermoFisher Scientific). The material was subjected to 25 passes in a Wheaton dounce tissue grinder (ThermoFisher Scientific) on ice before the addition of 200 µl chilled extraction buffer containing 0.5% Triton-X-100 per 100 mg of embryos. Following gentle mixing, samples were incubated on ice for 5 min and passed through a 23G syringe five times before clarification by two centrifugation steps (each 5 min at 3000 x *g*). 350 µl aliquots of clarified extract were incubated with 20 units Recombinant RNase Inhibitor (Promega, Madison, WI) and either 20 µl of a 6.7 µg/µl solution of unlabelled *hairy* RNA in dH$_2$O or 20 µl dH$_2$O for 30 min at 4°C. Magnetic beads coupled to GFP-binding protein (GFP-Trap MA (Chromotek, Martinsried, Germany)) were washed twice in PBS, followed by blocking of non-specific interaction sites with 1 mg ml$^{-1}$ casein in PBS for 30 min at 4°C. After two washes of the beads in extraction buffer, the equivalent of 30 µl of initial bead slurry was mixed with the embryo extracts with or without *hairy* RNA. Following a 2 hr 30 min incubation at 4°C, beads were washed fives times for 1 min in extraction buffer containing 0.05% Triton-X-100 (three washes in 400 µl of buffer and two washes in 1 ml buffer). Proteins and RNA-protein complexes were eluted from the beads by the addition of 60 µl 1 x lithium dodecyl sulphate (LDS) buffer (ThermoFisher Scientific)/50 mM DTT and incubation at 80°C for 10 min.

Following electrophoresis and blotting onto PVDF membranes, proteins were detected using the following primary antibodies: mouse $\alpha$-GFP (mix of clones 7.1 and 13.1 (Sigma-Aldrich; RRID:AB_390913); diluted 1:1000); mouse $\alpha$-Dhc (clone 2C11-C [*Sharp et al., 2000*]; RRID:AB_2091523) (provided by the Developmental Studies Hybridoma Bank (University of Iowa, Iowa, IA) and diluted 1:1000) and rabbit $\alpha$-p150-C-term ([*Kim et al., 2007*]; provided by V. Gelfand, Northwestern University; diluted 1:10,000). Secondary antibodies were conjugated to horseradish peroxidase, with signal detected using the ECL Prime system (GE Healthcare) and Super RX-N medical X-ray film (FUJIFILM, Bedford, UK).

## Analytical ultracentrifugation

Duplicate independent preparations of 1 mg ml$^{-1}$ Egl/BICD2 (2.4 µM assuming two Egl molecules and a single BICD2 dimer per complex) in GF150 buffer (using 5 mM TCEP instead of 5 mM DTT) in the presence or absence of a 10-fold molar excess of *ILS* RNA were pre-incubated on ice for at least 1 hr and subsequently diluted in GF150 (TCEP) to yield three samples with volumes of 110 µl and protein concentrations of 1, 0.33, and 0.11 mg ml$^{-1}$. These samples were loaded in 12 mm six-sector cells and subjected to equilibrium sedimentation in an An50Ti rotor using an Optima XL-I analytical ultracentrifuge (Beckmann) at 3200, 5600, and 10,000 rpm until equilibrium was reached at 4°C. At each speed, comparison of several scans was used to judge whether equilibrium had been reached. Data were processed and analysed using SEDPHAT 13b ([*Schuck, 2003*]; RRID:SCR_016254) and plotted with GUSSI ([*Brautigam, 2015*]; RRID:SCR_014962). The partial-specific volumes (v-bar),

solvent density and viscosity were calculated using Sednterp (T. Laue, University of New Hampshire; RRID:SCR_016253).

## SEC-MALS

Samples of BICD2, Egl/BICD2 and Egl/DmBicD were resolved on a Superdex 200 HR10/300 analytical gel filtration column (GE Healthcare) at 0.5 ml min$^{-1}$ in GF150 buffer (using 5 mM TCEP instead of 5 mM DTT), GF75 buffer (contains 75 mM KCl with 5 mM DTT), or GF50 buffer (contains 50 mM KCl with 5 mM TCEP). All measurements for BICD2 and Egl/BICD2 were made at room temperature, whereas the relative instability of the Egl/DmBicD complex required measurements be made at 4°C. Where indicated, *ILS* RNA was added at a 10-fold molar excess over Egl/BICD2 or Egl/DmBicD (based on an operational assumption of two Egl molecules and a dimer of the BicD protein per complex) and incubated on ice for 1 hr prior to injection on the column. Samples lacking RNA were subjected to the same incubation. Following SEC fractionation, eluted protein was detected on a Wyatt Heleos II 18 angle light scattering instrument coupled to a Wyatt Optilab rEX online refractive index detector in a standard SEC-MALS format. Heleos detector 12 at 99° was replaced with Wyatt's QELS detector for on-line dynamic light scattering measurements. Protein concentration was determined from the excess differential refractive index based on 0.186 RI increment for 1 g ml$^{-1}$ protein solution. Concentrations and observed scattered intensities at each point in the chromatograms were used to calculate the absolute molecular mass from the intercept of the Debye plot, using Zimm's model as implemented in ASTRA software (Wyatt; RRID:SCR_016255). Fractions were analysed by gel electrophoresis and staining with SYPRO Ruby (Lonza, Cambridge, UK) or Coomassie Instant Blue according to the manufacturer's instructions.

## Image analysis and statistics

Kymographs were generated and analysed manually using FIJI ([*Schindelin et al., 2012*]; RRID:SCR_002285). Typically, three independent chambers were imaged using protein complexes from at least two independent assembly reactions for each experimental condition. The positions of microtubules were determined by the fluorescent tubulin signal or a projection of RNA/protein signals over the course of the movie. From each of these chambers, 5–10 microtubules were typically selected for analysis with preference given to those that were longer and better isolated from adjacent microtubules. No power analysis was used to determine sample size. Instead the sample size was chosen to allow the identification of a range of effect sizes. To avoid the risk of subconscious bias, microtubules were selected before visualising the motile properties of complexes on them.

Interactions of fluorescently labelled proteins and RNA with microtubules were scored as binding events if they were ≥1.5 s (three frames) in duration and as processive events if they achieved predominantly minus end displacement >500 nm (five pixels) without significant diffusive behaviour. These parameters were chosen in advance of image acquisition following discussion within the team and all particles that fulfilled the criteria were analysed. As described previously (*Schlager et al., 2014*), some motile complexes changed velocity during a run, leading us to calculate velocities of individual constant-velocity segments. Run lengths were calculated from the total displacement of individual particles regardless of changes in velocity or pauses. For both velocity and run length calculations, only particles for which the entire run was observed or those with runs beginning >5 µm from the microtubule minus end were considered. When velocities and run lengths were calculated in the presence of RNA, only those complexes clearly associated with RNA were analysed. Although plots of 1 - cumulative frequency for run lengths were fitted to a one-phase exponential decay for visualisation purposes, statistical comparison of run lengths were performed on unfitted data. For *Figures 1D*, *2C* and *5B*, 'background' RNA binding was quantified by generating kymographs from random microtubule-free regions of the cover slip of lengths equal to the median microtubule length of those used for analysis. For illustrative purposes, the movie and kymographs in the figures had background subtracted in FIJI with a rolling ball radius of 50 pixels. All quantitative analysis was performed on the raw data. The gel analysis tools of FIJI were used to quantify the SYPRO Ruby signal in background-subtracted images (rolling ball radius of 50 pixels).

Statistical analyses, curve fitting, and data plotting were performed using Prism 7.0b (GraphPad; RRID:SCR_002798). A two-tailed Student's *t*-test or a two-tailed Welch's *t*-test was used when comparing two groups where a Gaussian data distribution was expected, with the latter test employed

in cases of unequal variance. A Mann-Whitney test was used to compare two groups with non-Gaussian data distributions. An ANOVA test with Dunnett's correction was used for multiple comparisons.

## Acknowledgements

We are very grateful to members of the Bullock lab, Carter lab and microscopy facility at MRC-LMB for advice and support, Andrew Carter for comments on the manuscript, Jianguo Shi for maintenance and propagation of Sf9 insect cells, Thomas Sladewski and Kathleen Trybus for sharing unpublished results, and Vladimir Gelfand for the p150 antibody. This work was supported by the UK Medical Research Council (file reference number MC_U105178790).

## Additional information

### Competing interests

Carly I Dix: Carly I Dix is currently affiliated with AstraZeneca Discovery Sciences. The research was conducted when the author was still at the MRC Laboratory of Molecular Biology. The author has no other financial interests to declare. Ha Thi Hoang: Ha Thi Hoang is currently affiliated with MicroInventa Limited. The research was conducted when the author was still at the MRC Laboratory of Molecular Biology. The author has no other financial interests to declare. The other authors declare that no competing interests exist.

### Funding

| Funder | Grant reference number | Author |
|---|---|---|
| Medical Research Council | MC_U105178790 | Mark A McClintock<br>Carly I Dix<br>Christopher M Johnson<br>Stephen H McLaughlin<br>Rory J Maizels<br>Ha Thi Hoang<br>Simon L Bullock |

The funders had no role in study design, data collection and interpretation, or the decision to submit the work for publication.

### Author contributions

Mark A McClintock, Conceptualization, Resources, Data curation, Formal analysis, Validation, Investigation, Methodology, Writing—original draft, Project administration, Writing—review and editing; Carly I Dix, Conceptualization, Resources, Investigation, Methodology; Christopher M Johnson, Stephen H McLaughlin, Formal analysis, Investigation, Methodology; Rory J Maizels, Resources, Investigation; Ha Thi Hoang, Investigation, Methodology; Simon L Bullock, Conceptualization, Supervision, Funding acquisition, Investigation, Writing—original draft, Project administration, Writing—review and editing

### Author ORCIDs

Stephen H McLaughlin (iD) http://orcid.org/0000-0001-9135-6253
Simon L Bullock (iD) http://orcid.org/0000-0001-9491-4548

### Decision letter and Author response

Decision letter https://doi.org/10.7554/eLife.36312.055
Author response https://doi.org/10.7554/eLife.36312.056

## Additional files

### Supplementary files

• Supplementary file 1. Expected labelling combinations of SNAP::BICD2 or Egl::SNAP if there are two copies of the SNAP-tagged polypeptide per complex.
DOI: https://doi.org/10.7554/eLife.36312.050

• Supplementary file 2. Observed outcome of dual labelling experiments with SNAP::BICD2 or Egl:: SNAP vs expectation for two SNAP-tagged polypeptides per complex.
DOI: https://doi.org/10.7554/eLife.36312.051

• Supplementary file 3. Correction of Cy3-*hairy* + Cy5 *hairy* RNA mixing results (*Figure 7G*) for the proportion of RNA molecules containing no dye.
DOI: https://doi.org/10.7554/eLife.36312.052

• Transparent reporting form
DOI: https://doi.org/10.7554/eLife.36312.053

### Data availability

All data generated or analysed during this study are included in the manuscript and supporting files.

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
