## [Decision Letter]

Thank you for submitting your article "RNA-directed activation of cytoplasmic dynein-1 in reconstituted transport RNPs" for consideration by *eLife*. Your article has been reviewed by three peer reviewers, and the evaluation has been overseen by Andrea Musacchio as the Senior and Reviewing Editor. The reviewers have opted to remain anonymous.

The reviewers have discussed the reviews with one another and the Reviewing Editor has drafted this decision to help you prepare a revised submission.

Summary:

The manuscript by McClintock and coworkers reports the establishment of an in vitro system to study dynein-dependent mRNA transport mediated by the adaptor Bicaudal D and its binding partner Egalitarian (Egl), which interacts with a specific subset of mRNAs containing appropriate targeting sequences. Using purified proteins (including BicD from both *Drosophila* and mouse) and different mRNAs known to be transported in vivo, the authors define a minimal set of components required for RNP motility in vitro. They then systematically dissect the requirements for RNP assembly, which identifies a crucial and unexpected role for the mRNA cargo. A combination of single-molecule motility assays and rigorous biochemical analysis allows the authors to present an insightful model for motile RNP assembly, which postulates that overcoming the autoinhibited state of BicD requires the binding of two Egl proteins to a BicD dimer, and that the Egl-BicD interaction is stabilized by mRNA binding to Egl.

All three reviewers are excited by the manuscript and recognize that the results represent a very important advance in our mechanistic understanding of polarized mRNA transport.

Essential revisions:

Reviewer #3 raised a concern related to the stoichiometry of the RNA in the complex. After an open discussion of the manuscript and of the comments, Reviewers #1 and #2 agreed that this point is very important in terms of understanding the nature of the complex and that it requires further attention.

• Figure 6B shows that BICD2 and Egl are fully complexed under the conditions used for SEC-MALS (since the Egl/BICD2 peak does not show a BICD2-like shoulder). This suggests that the RNA (*ILS*) is not required to recruit (at least some) Egl to BICD2. However, the SDS-PAGE below shows that more Egl is present in the larger complexes when *ILS* is added. This raises a few questions:

1) How much more Egl is recruited to BICD2 in the presence of *ILS*? Given that the authors have all the components purified, they should be able to quantitate a gel like that shown in this figure and use standards to determine the stoichiometry. Is there a single Egl in the Egl/BICD2 complex with a second one recruited by the *ILS*?

2) Since the *ILS* is present in a 10-fold molar excess in these experiments, why do the authors think it does not compete in what appears to be an *ILS*-mediated recruitment of additional Egl to the Egl/BICD2 complex?

• The data in Figure 7 suggest that a significant fraction of the transport complexes contain two RNAs. Figure 7B-E show that about 40% of complexes contain BICD2s and Egls with different dyes. Since one would expect 50% of the dimers to have two colors, this means that 80% of the complexes have two copies of BICD2, or Egl. Figure 7F,G shows that only 14% of complexes have RNAs with different dyes, meaning that 28% of complexes have two RNAs. Given all the data shows elsewhere in the paper, we can assume that those 28% of complexes come from the 80% containing two BICD2s and Egl. Therefore, 35% (28/80% ) of motile complexes have two RNAs. This does not seem to me to be "substantially lower" than the 65% of complexes that would have one RNA and two BICD2s and Egls, as the authors state. The composition of the activated transport complex is a major point of this work and thus I think this stoichiometry needs to be worked out more convincingly.

• It would be useful to include scrambled *ILS* controls for the SE-AUC and SEC-MALS, or at least the SEC-MALS if the SE-AUC is more demanding experimentally. Right now there is nothing in these experiments to rule out non-specific binding.

• The distribution of velocities in Figure 3E, Figure 3—figure supplement 1 and Figure 5—figure supplement 1 all show increases in velocity in the presence of RNA. However, these distributions appear bimodal, with low and high velocity populations. It seems it would be very useful, from a mechanistic point of view, to establish whether the RNA-induced increase in velocity is due to a shift of transport complexes from the slower to the faster population, without changing the velocities of each population. Since the general observation that dynein is faster in the presence of the RNA would not change, this is not a weakness of the manuscript, but if additional analysis of the data could answer the question it would add to it. Could this be an indication of a second dynein being recruited to the transport complex?

---

## [Author Response]

Essential revisions:Reviewer #3 raised a concern related to the stoichiometry of the RNA in the complex. After an open discussion of the manuscript and of the comments, Reviewers #1 and #2 agreed that this point is very important in terms of understanding the nature of the complex and that it requires further attention.• Figure 6B shows that BICD2 and Egl are fully complexed under the conditions used for SEC-MALS (since the Egl/BICD2 peak does not show a BICD2-like shoulder). This suggests that the RNA (ILS) is not required to recruit (at least some) Egl to BICD2. However, the SDS-PAGE below shows that more Egl is present in the larger complexes when ILS is added. This raises a few questions:1) How much more Egl is recruited to BICD2 in the presence of ILS? Given that the authors have all the components purified, they should be able to quantitate a gel like that shown in this figure and use standards to determine the stoichiometry. Is there a single Egl in the Egl/BICD2 complex with a second one recruited by the ILS?

We agree with the interpretation that RNA is not essential for (at least some) Egl to bind BICD2 in the SEC-MALS regime in Figure 6B. This conclusion is supported by the mean masses of the complexes measured in the absence of RNA compared to the BICD2 only sample, as well as the gel images. It is unfortunate that we did not make it clear in our first submission that these data are not consistent with a fixed stoichiometry of a BICD2 dimer and a single Egl in the absence of RNA. Rather, the relatively small increase in mean molar mass of the peak fraction compared to the BICD2 only sample, together with the gel images, indicates a mixture of complexed Egl/BICD2 (of indeterminate stoichiometry) and a large excess of free BICD2 dimer. The homogeneous values for mean molar mass across this peak and the lack of a BICD2-like shoulder suggest rapid association and dissociation of the minority Egl population with BICD2 dimers.

Similarly, although there is a clear overall increase in the occupancy of BICD2 with Egl in Figure 6B when RNA is present, the data do not support a uniform recruitment of two Egl proteins per BICD2 dimer. The broad range of mean masses across the Egl/BICD2 peak indicates that our experimental conditions captured an equilibrating mixture of different constituent species, which is likely to include both single and double occupancy of Egl with BICD2. Since these species were not sufficiently resolved by SEC (presumably due to relatively small differences in size/shape and/or rapid exchange), the MALS analysis provided the abundance-weighted mean mass of all of the species throughout the peak rather than masses of discrete states of complexation. Similar situations have been observed in SEC-MALS analysis of self-associating systems (e.g. van Breugel et al., *eLife* 2014 (PMID: 24596152)).

We have now modified the sections of the Results describing our SEC-MALS experiments (from the last paragraph of the subsection “The RNA localisation signal stabilises the Egl/BicD complex”) and the Figure 6 legend to clarify the key points discussed above.

The major conclusion we draw from the SEC-MALS experiments is that RNA pushes existing equilibria in favour of the occupancy of the BICD2 dimer with Egl (as illustrated in Figure 8). An important prediction of this model is that the equilibria can be shifted by modulating the concentration of Egl/BICD2. We now include additional data from a SEC-MALS experiment with a series of Egl/BICD2 concentrations (Figure 6D), which show that this is indeed the case.

As the reviewer indicates, a quantitative representation of this shift can be derived from gel images of peak SEC-MALS fractions. Making these measurements highly quantitative would require standards of Egl alone, which we were unable to purify (as now documented in Figure 1—figure supplement 2A,B). We are, however, able to measure the intensity of Egl bands relative to BICD2 bands from the peak fractions in the presence and absence of the *ILS*, providing an estimate of the relative increase in Egl signal when the RNA is present. This analysis is included in the legend to Figure 6. To further demonstrate the RNA-mediated increase in Egl/BICD2 association, we also include gel images from a new SEC-MALS experiment with lower ionic strength buffer (Figure 6—figure supplement 4B). The quantitative comparison of Egl to BICD2 signals in the gels is included in the legend.

Although the upper limits of the mean molar masses of mixtures of Egl, BICD2 and *ILS* in the SECMALS data are compatible with a fraction of complexes containing two Egl molecules and a BICD2 dimer, the analysis of single transport complexes in Figure 7 is much better suited to analyse protein copy number (a point we now make in the first paragraph of the subsection “The copy numbers of RNA, Egl and BicD in active transport complexes”). We believe that the analysis of protein copy number in active transport complexes in Figure 7 has been strengthened significantly as a result of the feedback from the reviewers (please see below). We now also present evidence from SEC-MALS that free Egl is indeed monomeric (Figure 6—figure supplement 3), which helps clarify possible mechanisms of RNA-mediated assembly of active adaptor protein complexes.

2) Since the ILS is present in a 10-fold molar excess in these experiments, why do the authors think it does not compete in what appears to be an ILS-mediated recruitment of additional Egl to the Egl/BICD2 complex?

As our data suggest dynamic association/dissociation of Egl and BICD2 it seems likely that the RNA-free Egl that is bound to BICD2 will be replaced with *ILS*-bound Egl. However, RNA-free and RNA-bound Egl complexes cannot be discriminated in our experiments.

• The data in Figure 7 suggest that a significant fraction of the transport complexes contain two RNAs. Figure 7B-E show that about 40% of complexes contain BICD2s and Egls with different dyes. Since one would expect 50% of the dimers to have two colors, this means that 80% of the complexes have two copies of BICD2, or Egl. Figure 7F,G shows that only 14% of complexes have RNAs with different dyes, meaning that 28% of complexes have two RNAs. Given all the data shows elsewhere in the paper, we can assume that those 28% of complexes come from the 80% containing two BICD2s and Egl. Therefore, 35% (28/80% ) of motile complexes have two RNAs. This does not seem to me to be "substantially lower" than the 65% of complexes that would have one RNA and two BICD2s and Egls, as the authors state. The composition of the activated transport complex is a major point of this work and thus I think this stoichiometry needs to be worked out more convincingly.

We regret that we did not adequately explain the implications of a small fraction of unlabelled protein for the interpretation of this experiment. As stated only in the Materials and methods, we achieved 90% of SNAP polypeptides labelled with a dye with our optimised procedures. Thus, the expected fraction of complexes that are dual-labelled assuming an obligate dimer is 0.405 ((0.9x0.9)/2). Our data therefore fit very well with there being two Egl polypeptides and two BICD2 polypeptides in almost all active transport complexes. We now explain this point in the Results (subsection “The copy numbers of RNA, Egl and BicD in active transport complexes”, second and third paragraphs), and include tables illustrating the calculations (Supplementary file 1 and 2).

To more accurately assess the fraction of transported complexes that contain two RNAs in the experiments documented in Figure 7F,G, we have now calculated the labelling efficiency of the labelled *hairy* RNA samples. Correcting for these values, our data indicate that 30% of motile complexes have two RNAs and 70% have one RNA (summarized in the Results, in the fourth paragraph of the subsection “The copy numbers of RNA, Egl and BicD in active transport complexes”, with supporting calculations in Supplementary file 3). We think that this is a substantial difference but do not make this point explicitly in light of the reviewer’s comment. The key conclusion is that transport of a single mRNA molecule by a complex containing two Egl molecules is not only possible but occurs frequently. This conclusion is consistent with our previous work showing that a single RNA can be transported by the native complex in an extract-based motility assay (Amrute-Nayak and Bullock, 2012; Soundararajan and Bullock, 2014) (cited in the aforementioned paragraph).

We have also clarified the basis of the association of two RNA molecules with a single transport complex by performing dual colour analysis of a mixture of Cy3-*hairy* and Cy5-*hairy* in the absence of the proteins. We find a very low incidence of co-localisation of these RNAs. This result indicates that in our motility assay, in which the RNA is present in a large molar excess, the presence of a second RNA in a subset of complexes is due to association with a protein(s) in the complex. This interaction was presumably blocked in our previous studies with extracts, as we always observed one RNA per transport complex. This observation calls into the question the physiological relevance of the binding of an additional RNA molecule. We have further analysed the data presented in Figure 7F and G to assess what affect, if any, the presence of multiple RNAs has on motility. Although there is a small effect on run length, the velocity distribution of complexes containing both Cy3-*hairy* and Cy5-*hairy* is very similar to that of complexes with signal from a single fluorophore. Thus, the presence of an additional RNA does not have substantial functional consequences for transport. We have included these analyses as Figure 7—figure supplement 2 and 3 and have revised the manuscript accordingly (see the aforementioned paragraph).

We believe that the new analyses strengthen significantly the section of the manuscript on the composition of active transport complexes.

• It would be useful to include scrambled ILS controls for the SE-AUC and SEC-MALS, or at least the SEC-MALS if the SE-AUC is more demanding experimentally. Right now there is nothing in these experiments to rule out non-specific binding.

The SE-AUC experiments require large amounts of protein, instrument time and analysis, which would indeed make this experiment very demanding. We have therefore taken up the option of performing this specificity control in the SEC-MALS format. We show in a new panel (Figure 6C) that the scrambled RNA elicits a relatively small shift in molar mass of Egl/BICD2 compared to the *ILS*. Our data indicate there is some interaction of the mutant RNA, which is in a 10-fold molar excess, with Egl/BICD2. This does not surprise us based on previous evidence that Egl is not a highly selective RNA-binding protein (Bullock et al., 2006; Dienstbier et al., 2009; Dix et al., 2013) (subsection “The RNA localisation signal stabilises the Egl/BicD complex”, last paragraph).

• The distribution of velocities in Figure 3E, Figure 3—figure supplement 1 and Figure 5—figure supplement 1 all show increases in velocity in the presence of RNA. However, these distributions appear bimodal, with low and high velocity populations. It seems it would be very useful, from a mechanistic point of view, to establish whether the RNA-induced increase in velocity is due to a shift of transport complexes from the slower to the faster population, without changing the velocities of each population. Since the general observation that dynein is faster in the presence of the RNA would not change, this is not a weakness of the manuscript, but if additional analysis of the data could answer the question it would add to it. Could this be an indication of a second dynein being recruited to the transport complex?

As suggested, we have performed additional analysis of the velocity distributions. Although it is possible to fit two Gaussian distributions to the data, we do not find the fits compelling. We therefore think it would not be appropriate to include them in the manuscript. We intend to investigate the potential role of dynein copy number in RNA transport in a separate study. We draw attention to the possible influence of dynein copy number in the Discussion (sixth paragraph).